# Stochastic tuning of gene expression enables cellular adaptation in the absence of pre-existing regulatory circuitry

Lydia Freddolino[1,2†], Jamie Yang[1,2], Amir Momen-Roknabadi[1,2], Saeed Tavazoie[1,2]*

[1]Department of Systems Biology, Columbia University, New York City, United States; [2]Department of Biochemistry and Molecular Biophysics, Columbia University, New York City, United States

**Abstract** Cells adapt to familiar changes in their environment by activating predefined regulatory programs that establish adaptive gene expression states. These hard-wired pathways, however, may be inadequate for adaptation to environments never encountered before. Here, we reveal evidence for an alternative mode of gene regulation that enables adaptation to adverse conditions without relying on external sensory information or genetically predetermined *cis*-regulation. Instead, individual genes achieve optimal expression levels through a stochastic search for improved fitness. By focusing on improving the overall health of the cell, the proposed stochastic tuning mechanism discovers global gene expression states that are fundamentally new and yet optimized for novel environments. We provide experimental evidence for stochastic tuning in the adaptation of *Saccharomyces cerevisiae* to laboratory-engineered environments that are foreign to its native gene-regulatory network. Stochastic tuning operates locally at individual gene promoters, and its efficacy is modulated by perturbations to chromatin modification machinery.
DOI: https://doi.org/10.7554/eLife.31867.001

*For correspondence:
st2744@columbia.edu

Present address: †Department of Biological Chemistry, University of Michigan Medical School, Ann Arbor, United States

**Competing interests:** The authors declare that no competing interests exist.

## Introduction

The capacity to adapt to changes in the external environment is a defining feature of living systems. Cells can rapidly adapt to familiar changes that are commonly encountered in their native habitat by sensing the parameters of the environment and engaging dedicated regulatory networks that have evolved to establish adaptive gene expression states (*Jacob and Monod, 1961*; *Thieffry et al., 1998*). However, dedicated sensory, signaling, and regulatory networks become inadequate, or even detrimental, when cells are exposed to unfamiliar environments that are foreign to their evolutionary history (*Tagkopoulos et al., 2008*). In principle, at least one gene expression state that maximizes the health/fitness of the cell always exists, despite the inability of the native regulatory network to establish such a state. This is true because under any conceivable environment, the activities of some genes are beneficial, whereas those of others are futile or even actively detrimental (*Jacob and Monod, 1961*; *Tagkopoulos et al., 2008*; *Hottes et al., 2013*). In fact, if the initial fitness defect is not lethal, a population of cells may slowly adapt to an unfamiliar environment through the accumulation of genetic mutations that rewire regulatory networks, thereby achieving more optimal gene expression states (*Tagkopoulos et al., 2008*; *Applebee et al., 2008*; *Philippe et al., 2007*; *Goodarzi et al., 2010*; *Tenaillon et al., 2012*; *Rodríguez-Verdugo et al., 2016*; *Blount et al., 2012*; *Van Hofwegen et al., 2016*; *Damkiær et al., 2013*).

**eLife digest** To survive, cells have to adapt to changes in their environment. Organisms can do so by constantly modifying the expression of their genes. For example, bacteria exposed to high temperatures turn on heat-shock genes to help them cope.

Responses to familiar environmental changes take place thanks to specific, hard-wired molecular pathways. These transmit external signals to transcription factors, proteins that can bind DNA near a gene to regulate its expression. Yet, such established responses may not exist for stressful conditions that cells have never encountered during their evolutionary history. In this case, how can organisms adjust which genes to express, and at what levels?

Here, Freddolino et al. theorize that, in a new environment, individual genes can randomly increase or decrease their level of expression. If a change ends up being good for the survival of the cell, it is further reinforced. This 'stochastic tuning' would allow organisms to find the optimal levels of gene expression without using genetically predetermined pathways that involve transcription factors.

Mathematical simulations suggest that this mechanism can improve the growth and survival of a cell in a new environment. Diverse experiments demonstrate that a phenomenon consistent with stochastic tuning occurs in yeasts. The organisms are genetically modified so that their transcription factors can no longer activate *URA3*, a gene required to grow in conditions lacking a chemical called uracil. Yet, these altered yeast cells still manage to boost their *URA3* expression in a uracil-free environment.

Stochastic tuning could thus work alongside other types of conventional gene regulation to help cells adapt to new and challenging living conditions. For instance, this may be how cancerous cells survive and thrive when facing chemotherapy drugs.

DOI: https://doi.org/10.7554/eLife.31867.002

## Results

### Adaptation through fitness-driven stochastic optimization of gene expression

In this work we speculate whether cells have evolved alternative strategies for finding adaptive gene expression states, on more physiological timescales, without relying on their hard-coded sensory and regulatory systems. Since the perception of the external world may be of limited value under unfamiliar conditions, perhaps a more effective strategy would be to focus on maximizing the internal health of the cell—without regard to the specific parameters of the outside world. This would be a challenging strategy, as every gene in the genome would need to independently reach the expression level that maximizes the overall health of the cell, and these expression levels could vary significantly from condition to condition. In particular, we asked whether individual genes could, in principle, carry out a search process equivalent to gradient descent (*Cauchy, 1847*), where the health consequence of stochastic alterations in gene expression could gradually tune the expression of individual genes towards a level that is optimal for internal health. We reasoned that such an optimization process would require the existence of: (1) a source of stochastic transitions in gene expression; (2) the ability of local chromatin to maintain a record of recent changes in transcription; and (3) a central metabolic hub that integrates diverse parameters of intracellular health and continuously broadcasts whether the overall health of the cell is improving or deteriorating. In fact, we find that the foundations for meeting these requirements are already present in eukaryotic cells: (1) The expression of many genes is dominated by noisy bursts of transcription—a widespread phenomenon of largely unknown functional significance (*Sanchez and Golding, 2013*; *Raj and van Oudenaarden, 2008*; *Blake et al., 2006*; *Raser and O'Shea, 2005*; *Elowitz et al., 2002*); (2) Co-transcriptional histone modification can modify eukaryotic chromatin in promoters and gene bodies, establishing a short-term memory of recent transcriptional events (*Li et al., 2007*; *Rando and Winston, 2012*); and (3) Global integrators of cell health have evolved in eukaryotes. A classic example is the mTOR pathway, which integrates a vast array of intracellular parameters reflecting nutrient availability, energy,

and the presence of diverse stresses (*Conrad et al., 2014*; *González and Hall, 2017*; *Albert and Hall, 2015*; *Saxton and Sabatini, 2017*).

With the necessary components for gradient-based optimization of gene expression in place (*Figure 1A*), the promoter of each gene would be able to conduct a simple search process that culminates in finding the expression level that maximizes the overall health of the cell: if global fitness/health is increasing *and* there was a previous increase in transcriptional output (representing larger or more frequent transcriptional bursts), the promoter further increases its transcriptional activity (*Figure 1B*). If fitness is decreasing *and* there was a previous increase in transcriptional activity, the promoter decreases its transcriptional output. Transcriptional output is altered in the opposite direction in the event that there was a previous decrease in transcriptional output. For each gene, this tuning process can be expressed as: $\Delta E_t = k \cdot sgn(\Delta F_t \cdot \Delta E_{t-1}) + \eta$ (see *Figures 1* and *2A*); here, $E$ denotes the vector of gene-level transcription rates, $F$ the current fitness/health of the cell, $k$ is a proportionality constant, $\eta$ a noise term, and *sgn* is a function yielding $-1$ if its argument is negative, 0 if its argument is zero, and $+1$ if its argument is positive. One can easily see how the process described here can tune the optimal expression of a single gene. What is remarkable, however, is the ability of this hypothetical stochastic tuning process to find near-optimal gene-expression states for a system with thousands of genes. As can be seen in the simulations presented in *Figure 2*, this is achieved through a fitness-directed stochastic search culminating in individual genes reaching specific gene expression levels that maximize the health/fitness of the cell. Such a stochastic tuning

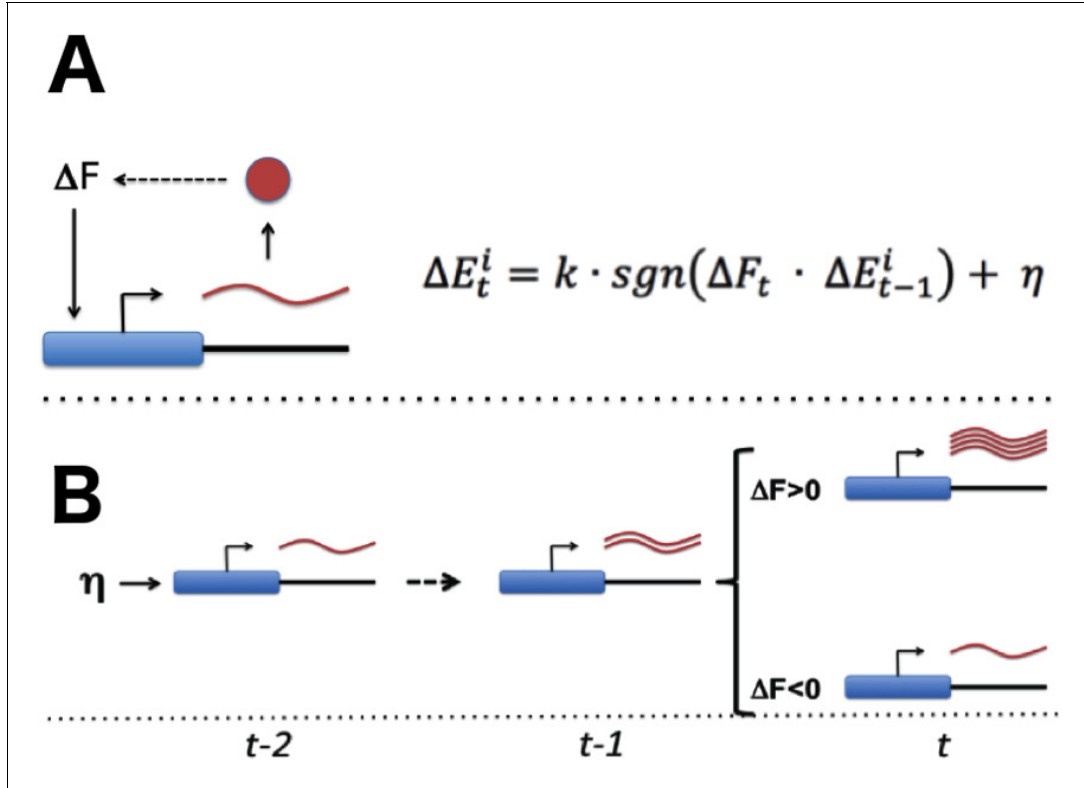

**Figure 1.** Stochastic tuning of gene expression by fitness optimization at gene promoters. (**A**) Each gene contains a noisy expression apparatus with noise amplitude η that allows exploration of a range of transcriptional activities. Each transcription apparatus also maintains a record of its previous change in transcriptional activity ($\Delta E_{t-1}$). The change in transcriptional activity has the potential to contribute to a change in global health ($\Delta F_t$) through the downstream effect of the gene product's activity (likely through a multi-step pathway; for example, the biosynthesis of a metabolite that is limiting for growth). A global metabolic integrator can transduce this change in health/fitness to every gene's expression apparatus. At any point in time, the expression apparatus executes a change in transcriptional activity ($\Delta E_t$) proportional (k) to the sign (*sgn*) of the product of $\Delta E_{t-1}$ and $\Delta F_t$ plus noise (η). (**B**) A simple example of this can be seen for a gene that experiences a random burst in transcriptional activity. If this leads to an increase in fitness the expression apparatus further increases transcriptional activity. Conversely, if there is a decrease in fitness, the expression apparatus decreases transcriptional activity.

DOI: https://doi.org/10.7554/eLife.31867.003

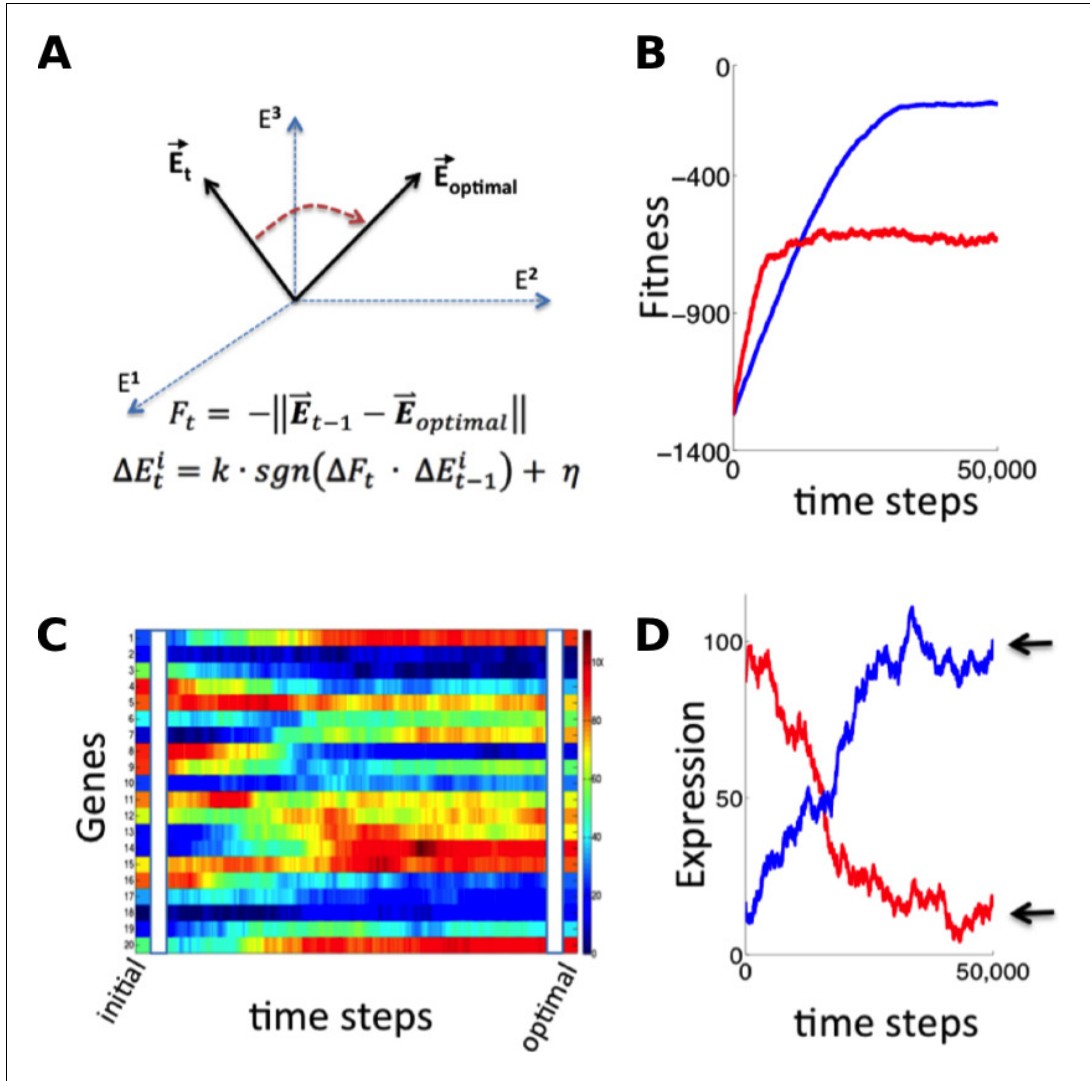

**Figure 2.** Simulation of fitness-directed stochastic tuning for a thousand-gene system. (A) Quantitative framework describing stochastic tuning. The transcriptional activity state of the genome is represented by the vector E, here schematically represented for a three-gene system. In any environment, there is an optimal transcriptional state vector ($E_{optimal}$) that yields maximum fitness. At any time (t), a cell with transcriptional activity state $E_t$ has global health/fitness ($F_t$) defined as the negative of the Euclidean distance between the immediately preceding transcriptional activity state $E_{t-1}$ and $E_{optimal}$. Each gene promoter (i) executes a change in transcriptional activity $\Delta E^i_t$ which has two components: (1) a step with magnitude of $k$ and sign ($sgn$) matching that of the product of the global change in fitness ($\Delta F_t$) experienced at time $t$ and the preceding change in transcriptional activity $\Delta E^i_{t-1}$, and (2) a noise component with a magnitude of $\eta$ and a random sign (+/-). (B) The stochastic tuning process moves the transcriptional activity state towards the optimum, resulting in increasing health/fitness over time. Simulated trajectories are shown for a 1,000-gene system with $k = 0.1$, $\eta = 0.1$ (blue); $k = 0.5$, $\eta = 0.5$ (red). (C) The time evolution of the transcriptional activity state vector as a system containing 1000 genes converges to optimal transcriptional activities through stochastic tuning. The temporal profiles of 20 representative genes are shown, starting from randomly assigned initial activities, and gradually converging to activities that are near optimal for fitness (using parameters corresponding to the blue curve in panel B). (D) Trajectories of two representative genes are shown for the same simulation as in panel C). Transcriptional activities start at randomly assigned initial values and gradually converge to near the optimum (arrows).

DOI: https://doi.org/10.7554/eLife.31867.004

mechanism would be highly valuable to free-living organisms, enabling them to optimize their global gene expression patterns to match the specific requirements of any environment in which their dedicated sensory and regulatory networks are inadequate or sub-optimal.

## Fitness-directed tuning of gene expression in yeast

Informed by the simulations above, we sought to test for evidence of stochastic tuning in the eukaryotic model organism *Saccharomyces cerevisiae*. We engineered conditions in which the expression of a single gene was required for growth, but for which no regulatory input existed to drive appropriate expression levels. This was achieved by using a yeast strain (BY4743) that lacks the URA3 gene, which is essential when cells are grown in the absence of uracil. We placed a chromosomally integrated copy of URA3 at a different locus under the control of a weak synthetic promoter, consisting primarily of a pseudorandom sequence. All recognizable binding sites for native transcription factors were removed from the generated promoter sequence (see Materials and Methods and *Supplementary file 1* for details), in an attempt to decouple it from any existing sensory and regulatory input. We henceforth refer to this synthetic promoter sequence as synprom (see *Supplementary file 1* for sequence). In the experiments described below, URA3 is typically tagged with a fluorescent fusion, either mRuby (*Kredel et al., 2009*) or a superfolder GFP (*Pédelacq et al., 2006*), and a copy of a mouse DHFR gene coupled to a different fluorescent protein is inserted at the same location on the sister chromosome to act as an internal control. A schematic of the insertion constructs is shown in *Figure 3A*. We also added the URA3 competitive antagonist 6-azauracil (6AU) to the media to control the threshold level of URA3 production required for growth. The growth condition, SC+glu-ura media, containing x μg/ml of 6AU, will henceforth be referred to as ura-/6AUx.

Even with the challenging and specific experimental layout described here, with growth highly dependent on URA3 expression, we expect that stochastic tuning might contribute to fitness through mechanisms acting in cis at the promoter driving URA3, those acting in trans through modulation of factors that (despite our best efforts) weakly affect the promoter driving URA3, and through tuning of unrelated pathways that benefit survival and growth in the –URA condition. Nevertheless, URA3 expression itself will clearly be the key driver of growth since it is the critical bottleneck for nucleotide biosynthesis in the absence of uracil supplementation.

To look for evidence of fitness-directed stochastic tuning, we tracked the colony formation of cells containing synprom-driven URA3 after plating on ura-/6AU15 plates. Lacking sufficient URA3 expression to overcome high 6AU levels, these non-growing cells would be expected to succumb to starvation and die. Remarkably, however, after prolonged incubation we observed apparently stochastic transitions to rapid growth, leading to the formation of macroscopic colonies over time (*Figure 3B*). We eventually observed colony formation by roughly one cell in $10^3$, a rate too high to be driven by mutation-driven adaptation in the absence of growth.

## Stochastic tuning of other synthetic and natural promoters

The synthetic promoter referred to as 'synprom' throughout the text is the combination of a pseudorandom sequence with a small natural promoter-proximal region taken from the SAM3 gene, with both stripped of all recognizable matches to known transcription factor binding sites (see Materials and Methods for details). We also tested all combinations of five other synthetic promoter sequences and one other promoter proximal region, enumerated in *Supplementary file 2*. As shown in *Figure 3—figure supplement 1*, four of the six synthetic promoters support stochastic tuning, and the ability of synprom5 (the purely artificial component of the synprom referred to in the remainder of the text; see *Supplementary file 2* for all synthetic promoter sequences) to undergo tuning remains even with a different promoter proximal region. These findings highlight the universality of the observed tuning phenomenon and minimize the possibility that our observations actually arise due to the presence of some residual sequence-specific transcription factor binding site present in synprom.

As shown in *Figure 3C*, we also observed similar tuning behavior for two high-noise natural promoters, $P_{HSP12}$ and $P_{RGI1}$ (*Tirosh et al., 2009*; *Tirosh et al., 2006*), indicating that stochastic tuning can function even when superimposed on naturally evolved regulatory sites. Across all promoters

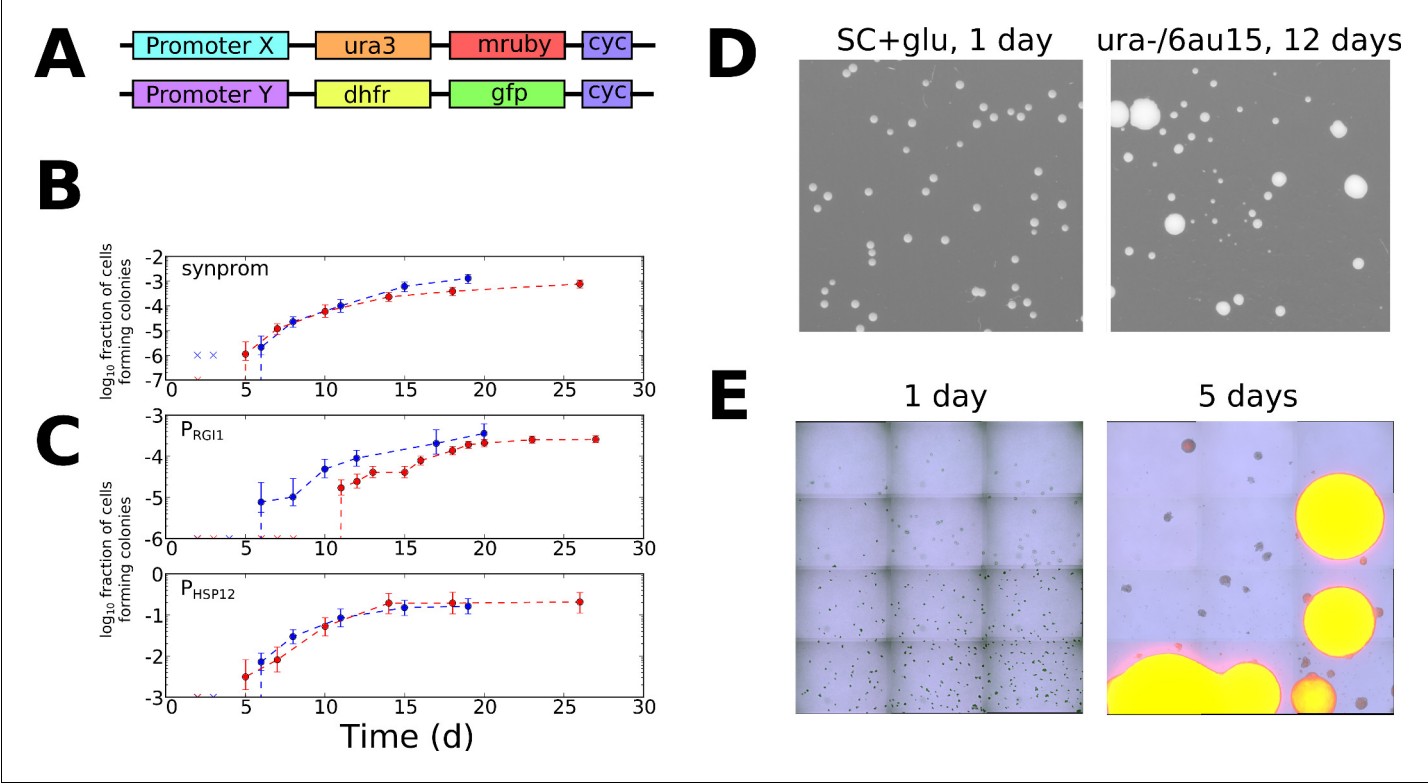

**Figure 3.** Stochastic tuning of yeast cells under uracil starvation. (A) Schematic of the constructs used in this study. All strains are diploid, containing similar insertions at the LEU2 locus of both copies of chromosome III. X is either a synthetic promoter (synprom) or a natural promoter ($P_{RGI1}$ or $P_{HSP12}$) unless otherwise noted, and Y is either the same promoter as X or is the strong constitutive promoter $P_{ADH1}$. 'cyc' indicates the well-characterized CYC1 transcriptional terminator (**Russo and Sherman, 1989**). (B) Stochastic colony formation on ura-/6AU15 plates for cells containing URA3-mRuby under control of synprom and DHFR-GFP under control of $P_{ADH1}$. Error bars show central 95% credible intervals; colors show biological replicates performed on different days. 'x' marks are shown at the bottom of the axis for days where zero visible colonies were present at all plated dilutions. Cells plated on SC+glu uniformly form visible colonies within 1–2 days. (C) As in panel B, but with URA3-mRuby controlled by $P_{RGI1}$ or $P_{HSP12}$ as indicated. (D) Images of colony growth on SC+glu and ura-/6AU15 plates taken at the specified number of days after plating (1 day for SC+glu, 12 days for ura-/6AU15). Growth of colonies is nearly uniform on SC+glu plates but shows non-uniform stochastic emergence on ura-/6AU15. *N.b.* the plated dilutions for the two plate types are not the same. URA3 expression for the experiment shown is controlled by $P_{HSP12}$, but similar behavior was observed for all promoters discussed here. (E) Early colony formation on ura-/6AU15 plates imaged by superimposed differential interference contrast and fluorescence microscopy. Cells contain $P_{HSP12}$-URA3-mRuby/$P_{ADH1}$-DHFR-GFP. Left panel: One day after plating. By this timepoint small, macroscopic colonies would have formed on SC+glu plates, but instead cells remain in microcolonies having undergone no more than three doublings. Right panel: Same plate as left, five days after plating. While most cells have not grown since the one-day timepoint, other cells having undergone successful tuning instead form larger colonies with URA3 expression sustained throughout them.

DOI: https://doi.org/10.7554/eLife.31867.005

The following figure supplement is available for figure 3:

**Figure supplement 1.** Stochastic colony formation rates for cells with URA3 driven by a variety of synthetic promoters.

DOI: https://doi.org/10.7554/eLife.31867.006

(natural and synthetic) tested here, the observed tuning rates, relative to the number of viable plated cells, varied from 1 in $10^1$ ($P_{HSP12}$) to 1 in $10^5$ (synprom5-arf1).

The apparently stochastic nature of colony formation in our experiments is reflected both in the steady emergence of colonies over the course of days or weeks (**Figure 3B–C** and **Figure 3—figure supplement 1**), and in the wide variance of colony sizes observed on ura-/6AU15 plates (**Figure 3D**). Microscopy revealed that cells remain quiescent for days before transitioning to URA3 expression and rapid growth, with a transition rate dependent on the choice of promoter (**Figure 3E**). Furthermore, the change that enables growth under the ura-/6AU15 condition must be passed from mother to daughter cells, as colonies expand from a few points of initiation instead of showing random division of cells throughout the microscopic field over time. While the presence of some deterministic

process, yielding colony formation over the observed timescales (dependent on the initial state of each cell), cannot be ruled out, a far simpler explanation for the observed phenomenon of a long lag followed by appearance of colonies over a wide range of times is that each cell independently undergoes a random process that can eventually lead to growth. We confirmed that the appearance of colonies is not simply due to aging of the plates; 6AU-containing plates which were pre-incubated for a week or longer prior to plating of cells showed no change in colony formation rates (data not shown).

## Fitness-directed tuning operates independently of conventional regulatory input and is transcriptionally driven

To provide further insights into the regulatory changes occurring during the onset of cell growth, we performed flow cytometry time courses on cells challenged by, and subsequently growing in, liquid ura-/6AU5 media, using cells with synprom-driven URA3-mRuby, and with a DHFR-GFP fusion driven by either the constitutive ADH1 promoter (*Figure 4A–B*) or synprom (*Figure 4C–D*) itself. The use of P$_{ADH1}$ to drive the second reporter allows us to control for extrinsic noise and global changes in gene expression, whereas coupling synprom to the non-beneficial DHFR-GFP fusion allows us to test whether the observed stochastic tuning is driven by any *trans*-acting input from some existing

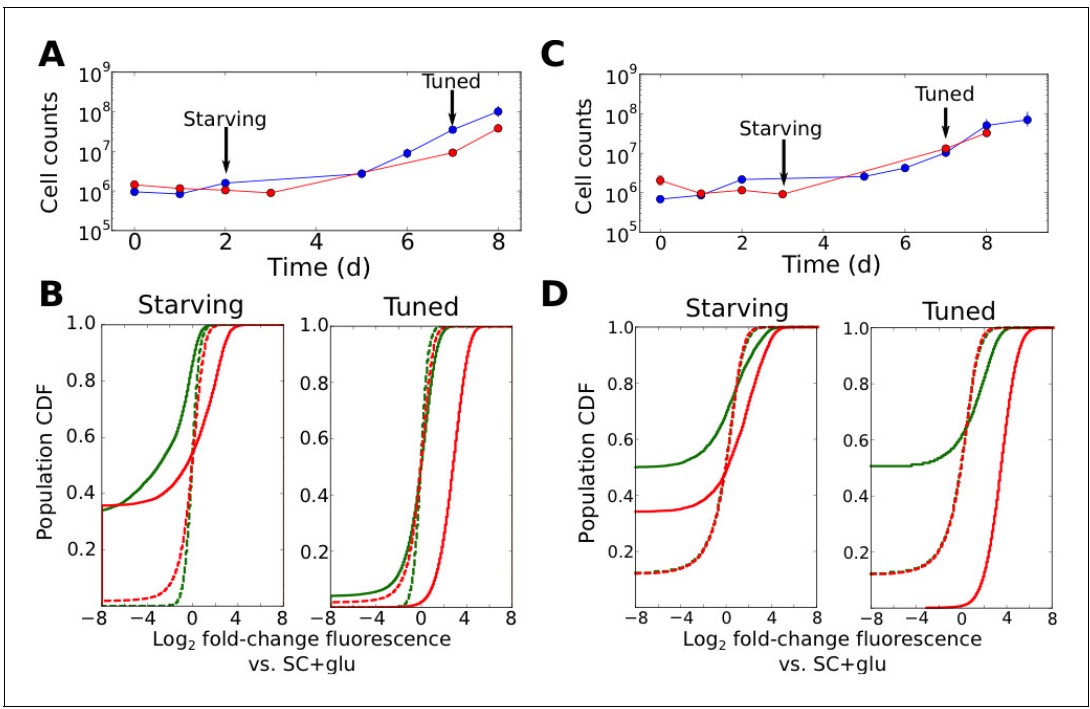

**Figure 4.** Tuning is both promoter- and allele-specific. (A) Cell counts for synprom-URA3-mRuby/P$_{ADH1}$-DHFR-GFP cells in liquid ura-/6AU5 media. Colors correspond to different biological replicates started on different days. Arrows indicate two timepoints from each strain for which fluorescence cumulative distribution functions (CDFs) are shown below. Error bars for cell counts show central 95% credible intervals. (B) Flow cytometry cumulative distributions of fluorescence levels for URA3-mRuby and DHFR-GFP during uracil starvation. In each CDF a given timepoint (solid line) is compared to the distribution present for cells in logarithmic growth in SC+glu (rich) media (dashed lines). The values shown are log$_2$ ratios to the median value of cells growing exponentially in SC+glu. GFP signals are shown in green and mRuby signals in red. (C) Analogous to A, but we consider cells where synprom drives both URA3-mRuby and DHFR-GFP. (D) Analogous to B, but for cells with synprom driving both URA3-mRuby and DHFR-GFP.
DOI: https://doi.org/10.7554/eLife.31867.007

The following figure supplements are available for figure 4:

**Figure supplement 1.** Promoter-specific stochastic tuning of URA3 expression by native promoters in *S. cerevisiae*.
DOI: https://doi.org/10.7554/eLife.31867.008

**Figure supplement 2.** Local tuning of URA3 expression.
DOI: https://doi.org/10.7554/eLife.31867.009

regulatory network or whether it is truly specific to the allele needed for growth, as required by our proposed tuning model.

Several patterns in the growth curves and flow cytometry data are immediately apparent. First, as with the agar-based growth discussed above, cells show a lag of at least 72 hr with undetectable growth, followed by the onset of steady growth until saturation. In the case of URA3-mRuby driven by synprom and DHFR-GFP by the constitutive promoter $P_{ADH1}$, URA3-mRuby fluorescence increases substantially in tandem with the onset of cell growth, and expression subsequently remains high until saturation; in contrast, DHFR-GFP signals do not even recover to their initial levels (*Figure 4B*; compare dashed and solid line distributions). This demonstrates that the URA3 induction resulting in growth is promoter-specific and does not simply reflect a general increase in protein expression. We observed qualitatively equivalent behavior when URA3 was driven by $P_{RGI1}$ or $P_{HSP12}$ (*Figure 4—figure supplement 1*). Even more strikingly, for cells with synprom driving both fluorescent fusions, we observed a specific enhancement of URA3-mRuby expression over that of DHFR-GFP (*Figure 4D*), showing that the transition to high URA3 expression is not only promoter-specific but allele-specific, and thus must be driven at least partly by changes occurring in cis at the specific locus whose expression is required for growth. As an additional test, we performed quantitative RT-PCR experiments to measure the ratio of URA3 and DHFR mRNA expression in tuned cells either in liquid ura-/6AU5 media or on ura-/6AU15 plates (see *Figure 4—figure supplement 2*). In both cases, we observed a substantial increase in the URA3:DHFR ratio in the tuned cells, indicating that the observed tuning occurs at least partly through a local *cis*-acting process at the locus required for growth (although we cannot rule out additional changes in other promoters that also contribute to survival and growth, which may account for the observed heterogeneity in expression levels between replicates). Consistent with our proposed tuning model, the allele-specific nature of the transcriptional induction supports a key role for a local tuning process that is independent of dedicated sensory and regulatory input.

## Varying the threshold level of URA3 required for growth shifts tuning from stochastic to deterministic

The presence of the competitive URA3 inhibitor 6-azauracil allows us to vary the threshold level of URA3 required for growth. Thus, it is instructive to consider how the concentration of 6AU may alter stochastic tuning behavior, both in the context of the computational model described above and in the actual behavior of the system. We made two crucial modifications to the numerical model employed in *Figure 2* to mimic our experimental setup. First, rather than having the entire gene expression profile begin far from the optimal point, we begin with all genes but one (representing URA3) at their optimal values, reflecting the fact that aside from the artificial stress of lacking appropriate URA3 regulation, the cells' native regulatory network can provide an appropriate response to ura-/6AU media. Second, we note that due to the presence of the competitive inhibitor 6AU, the URA3 in the cell will not even be able to contribute meaningfully to nucleotide biosynthesis (and thus impact the cell's health/fitness) until it passes a threshold level. Thus, the tuning term (*Figure 2A*) is not applied to the gene representing URA3 until after the concentration of URA3 passes a threshold. Aside from the modifications noted above, we model tuning in the ura-/6AU environment as we did for the general case in *Figure 2*, and in particular, the fitness effects of changing URA3 expression must compete with noisy gene expression from the other 999 genes in the model gene expression profile to impact the direction of tuning.

The resulting URA3 expression profiles during simulated tuning in the presence of low or high concentrations of 6AU are shown in *Figure 5A*. In the low 6AU case, the tuning mechanism pushes URA3 expression almost deterministically to its optimal (high) value, whereas in the presence of high 6AU, the URA3 expression level undergoes a random walk until expression becomes high enough to allow the tuning mechanism to 'sense' the gradient and drive the cells into a URA3+ state. The effects on tuning rates of varying the 6AU concentration are plotted in *Figure 5B*, where we observe that increasing 6AU concentrations both slow tuning and dramatically increase the variance in the amount of time required for each individual cell to reach a URA+ state. This is precisely the behavior observed experimentally with high 6AU concentrations (*Figure 3*). On the other hand, tuning in our experimental system switched from slow and stochastic to rapid and deterministic in the presence of low 6AU concentrations, with observable tuning occurring over the course of a few hours (*Figure 5C*). Importantly, the tuning process is confined to the URA3-mRuby allele, despite the fact

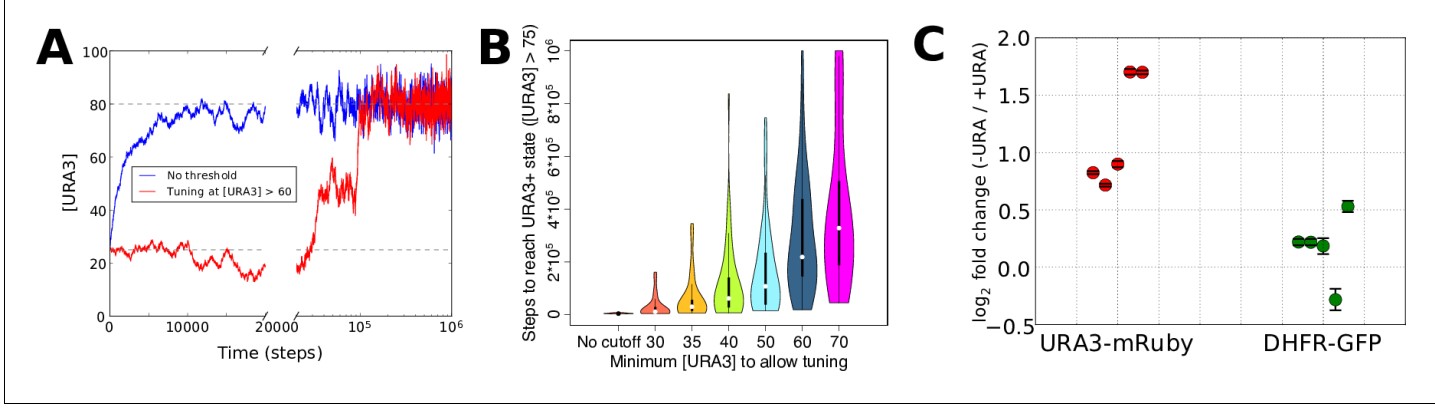

**Figure 5.** Numerical modeling and experimental validation of changes in tuning behavior as a function of 6AU concentration. We simulated the gene expression dynamics of cells containing URA3 under the control of a non-native promoter, when exposed to uracil-depletion stress with varying concentrations of the URA3 inhibitor 6AU. The model employed is equivalent to that in *Figure 2A*, with $k = 0.1$, $\eta = 0.1$, and the target expression profile equal to that for the case shown in *Figure 2B* except for the case of the gene corresponding to URA3, whose optimal value was set to 80. (**A**) Typical trajectories of URA3 expression levels for a cell in the presence of low (blue) or high (red) 6AU concentrations, which alter the minimum URA3 expression level at which fitness-directed stochastic tuning can occur. We show results for a starting URA3 level [URA3]=25, with optimal fitness occurring at [URA3]=80. The initial and optimal URA3 levels are shown as gray lines. (**B**) Violin plots of the distributions of the minimum time required to reach a URA3+state (defined as [URA3]>75) in the presence of increasing concentrations of 6AU (implemented as higher thresholds of URA3 required for stochastic tuning to become active). In each case distributions reflect 50 independent trajectories simulated at each 6AU level. (**C**) Experimental validation of model predictions. Cells were grown in liquid ura-/6AU1 media (-URA) for 3–4 hr and then had the expression of fluorescent reporter proteins compared (using flow cytometry) with those of the equivalent cells grown in SC+glu (+URA) over the same time period. Values show $\log_2$ fold changes from SC+glu to ura-/6AU1; error bars show bootstrap-based 95% confidence intervals. Biological replicates performed on different days are shown side by side; the order of replicates is matched for URA3-mRuby and DHFR-GFP.

DOI: https://doi.org/10.7554/eLife.31867.010

that DHFR-GFP is also being driven by the same synthetic promoter. This again demonstrates that the tuning process occurs independently of conventional gene regulation by dedicated sensory and regulatory input.

## Tuning dynamics at the single-cell level

We utilized time-lapse fluorescence microscopy to monitor the correspondence between expression of URA3-mRuby and cell division in $P_{HSP12}$-URA3-mRuby/$P_{ADH1}$-DHFR-GFP cells that initiated the tuning process. Consistent with our proposed tuning model, gene expression fluctuations that surpassed a threshold for alleviating the URA3 deficit were reinforced over long timescales and were

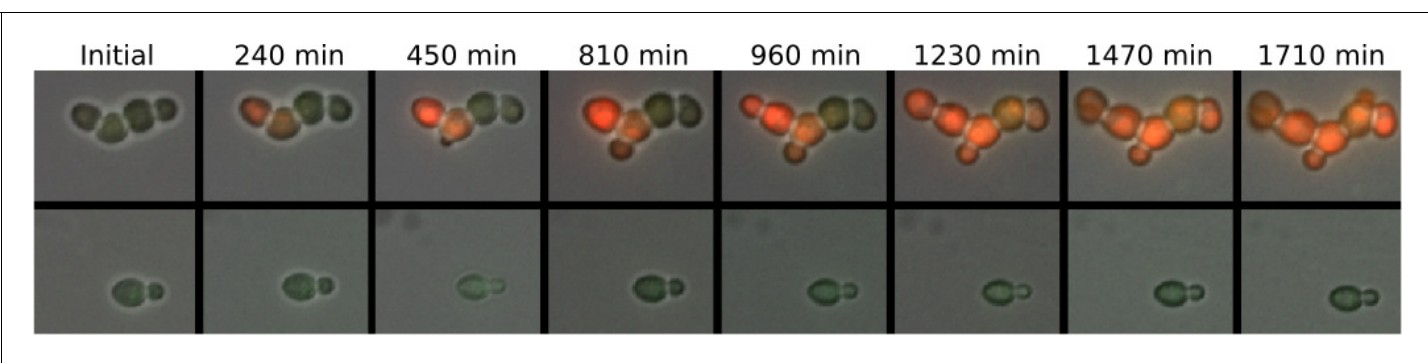

**Figure 6.** Sustained trans-generational inheritance of URA3-mRuby expression in tuned microcolonies. Shown are fluorescence microscopy time courses of microcolonies beginning after 12 hr of exposure to ura-/6AU5 media. A tuned colony is shown on top and a nearby untuned colony on the bottom. Fluorescence values are uniformly scaled but are not otherwise processed.

DOI: https://doi.org/10.7554/eLife.31867.011

sustained (inherited) across multiple generations as the tuned colony expanded (*Figure 6*). As expected, there is no accompanying increase in DHFR-GFP. Similar trajectories were observed for other tuning micro-colonies (*Figure 7A*). The apparently long autocorrelation time of URA3-mRuby fluctuations through the duration of a tuning trajectory is consistent with our proposed fitness feedback reinforcement mechanism. In order to quantitatively determine the timescale of gene expression fluctuations, also known as mixing times (*Sigal et al., 2006*), we utilized fluorescence-activated cell sorting (FACS) to sort a population of cells for the bottom 20%, the top 20%, and complete (mock-sorted) distribution of URA3-mRuby expression and measured the timescales over which the sorted fluorescence distributions converged to each other (*Figure 7—figure supplement 1*). For cells growing under uracil-replete conditions (SC+glu), we observed a relatively fast mixing time on the order of ~100 min (*Figure 7—figure supplement 1*; *Supplementary file 3*). On the other hand, cells starving in ura-/6AU10 media had mixing times that ranged from 400 to 1200 min (*Figure 7—figure supplement 1*; *Supplementary file 3*).

To determine the association of URA3-mRuby levels across generations with growth, we primed cells with 12 hr of exposure to ura-/6AU5 media and then tracked the division of tuned vs. untuned microcolonies of $P_{HSP12}$-URA3-mRuby/$P_{ADH1}$-DHFR-GFP cells over 24 hr time courses in ura-/6AU5 media. By comparing the fluorescence of cells that are about to divide with those that are not, we found that dividing cells have significantly higher levels of mRuby than non-dividing cells, whereas the separation was much smaller for GFP (*Figure 7C*). Furthermore, the URA3-mRuby levels within the tuning colony were highly heritable; as seen in *Figure 7D*, as the indicated colony tunes and grows, cells within that colony maintain a high-mRuby state through subsequent divisions, and even their internal rankings are mostly preserved. mRuby levels in other, non-tuned microcolonies are almost uniformly lower than cells in the tuned colony. The fitness-driven optimization component of our model (*Figure 1*) further predicts that fluorescence levels should not only be heritable, but also that cells will continue to increase URA3 expression (possibly noisily) until they reach either a local optimum fitness or some biological constraint on maximum gene expression. Consistent with our expectation, we observed that the ratio of mRuby to GFP levels (the latter of which is fused to a gene whose product is not needed for growth) became steadily higher in cell lineages that had been dividing for longer (*Figure 7E*). These observations demonstrate that the level of URA3 expression is correlated with fitness, is transmitted across several generations, and shows an ongoing upward trend in tuned cells over the course of time. That last finding is particularly important because a directed increase in URA3 once a lineage begins growing is predicted by our model for fitness-directed tuning, but cannot be explained by other competing hypotheses. The images and data shown in *Figure 7* were taken for colonies within a single field of view of a 40x objective to ensure internal consistency in illumination and normalization, but their behavior is representative of our observations across multiple such windows. (*e.g.*, *Figure 7—figure supplement 2*, panel A). Similar quantitative analysis from another experiment beginning directly from growth in SC+glu (instead of short-term pregrowth in ura-6AU media) is shown in *Figure 7—figure supplement 2*, panels B-D.

## Growth on –ura/6AU media does not arise from genetic mutations

It is crucial to exclude the possibility that genetic mutations underlie the observed tuning transition on –ura/6AU plates. The ongoing emergence of the tuned state in non-growing cells, over the course of many days, makes mutational mechanisms unlikely. In addition, as seen by microscopy (*Figure 3E*), no more than 1–3 cell divisions occur prior to the onset of sustained growth in a small fraction of cells.

Nevertheless, given the phenomenon of stress-induced mutagenesis in non-growing bacterial cells (*Al Mamun et al., 2012*), we wished to conclusively exclude any possibility of mutational mechanisms. To this end, we note that changes in URA3 expression occurring due to mutations should be stably heritable in the progeny of the tuned cells, which would not be expected to revert to a URA3 low state even after restoration of uracil in the media. To test the reversibility of the URA3 high state, we designed an experimental setup in which tuned colonies isolated from ura-/6AU plates were grown for varying numbers of passages in uracil-replete media (SC+glu including uracil) and then re-exposed to uracil starvation (*Figure 8—figure supplement 1*). If any genetic mutation were responsible for increasing URA3 expression in the tuned cells, the phenotype should be stable for many generations. On the other hand, stochastic tuning would predict that cells revert to a naïve state following sufficient growth in uracil-containing conditions, as they no longer benefit from URA3

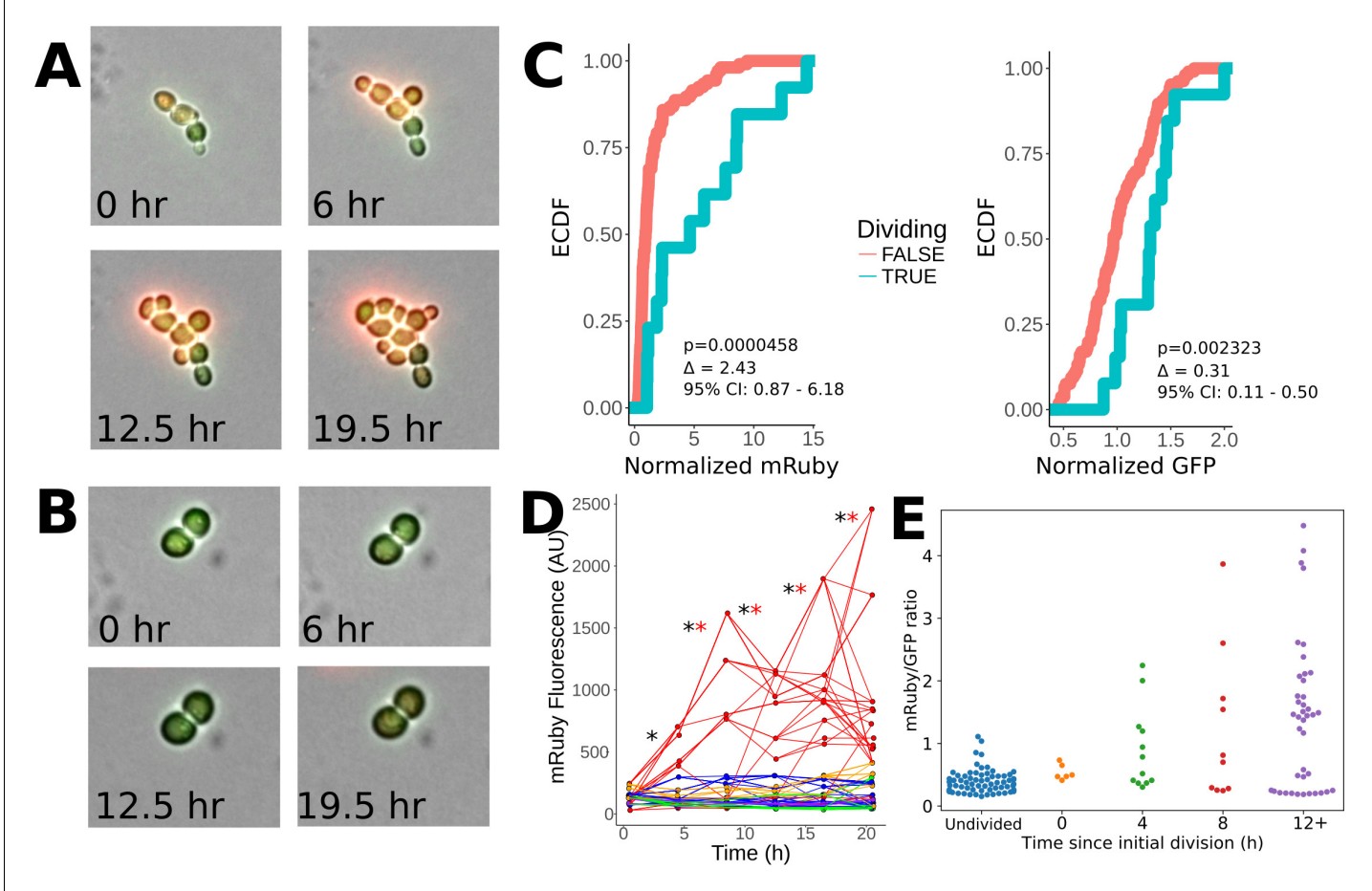

**Figure 7.** Heritability of elevated mRuby levels during tuning. (A) Formation of a microcolony over 24 hr of exposure to ura-/6AU5 media in P$_{HSP12}$-URA3-mRuby/P$_{ADH1}$-DHFR-GFP cells. GFP and mRuby are shown as transparent green and red overlays. (B) Snapshots equivalent to A) for a non-tuned colony in the same field of view. (C) Observed cumulative distributions (empirical cumulative distribution function; ECDF) of mRuby (left) and GFP (right) levels for cells that either do or do not divide in the timepoint following the measurement (analyzed in four-hour intervals). Values are pooled over all timepoints except the first, for five colonies growing in a single field of view. p-values arise from a Wilcoxon rank sum test applied to the shift between the non-dividing and dividing cells. Δ indicates a point estimate for the difference in fluorescence of the dividing vs. non-dividing cells, along with a 95% confidence interval (95% CI). Values shown are raw fluorescence normalized by the median value for all observations of each fluorescent protein; note the different x scales for mRuby vs. GFP. (D) Lineage traces showing long term propagation/inheritance of URA3-mRuby protein levels. At each specified timepoint, the average fluorescence of each cell is shown on the y axis, with lines connecting each cell to the cell(s) arising from it at the subsequent timepoint; thus, forks in the lines indicate cell division. Colors specify which of five microcolonies a given cell is a part of; only the red microcolony showed notable tuning over the course of the experiment. A black '*' is shown for each transition between adjacent timepoints for which the correlation of ranks between the timepoints in question is significant (p<0.05) using a Spearman correlation test, and a red '*' is shown for transitions where the same criterion holds considering only the rank ordering of cells in the red (tuned) colony (the colony shown in panel A). (E) Observed distribution of mRuby/GFP ratios depending on time elapsed since a lineage of cells began to divide. The x axis divides the cells up by the time (measured in four-hour intervals) that has elapsed since the first observed division event of an ancestor of that cell; 'Undivided' indicates cells in lineages that have not yet divided in the analyzed trajectory, and 0 hr denotes cells that will divide before the next analyzed snapshot. Note that points are plotted for each cell at each analyzed frame relative to its own growth history, and thus not all cells at a given x position necessarily arise from the same time point in the image series.

DOI: https://doi.org/10.7554/eLife.31867.012

The following figure supplements are available for figure 7:

**Figure supplement 1.** Mixing times of mRuby levels for growing (uracil-replete) and uracil-starved cells.
DOI: https://doi.org/10.7554/eLife.31867.013
**Figure supplement 2.** Heritability of elevated mRuby levels during early tuning.
DOI: https://doi.org/10.7554/eLife.31867.014

expression. As seen in *Figure 8A*, cells with synprom-driven URA3 show reversion toward the naïve colony formation rates upon growth in (uracil containing) SC+glu media, with recovery apparent

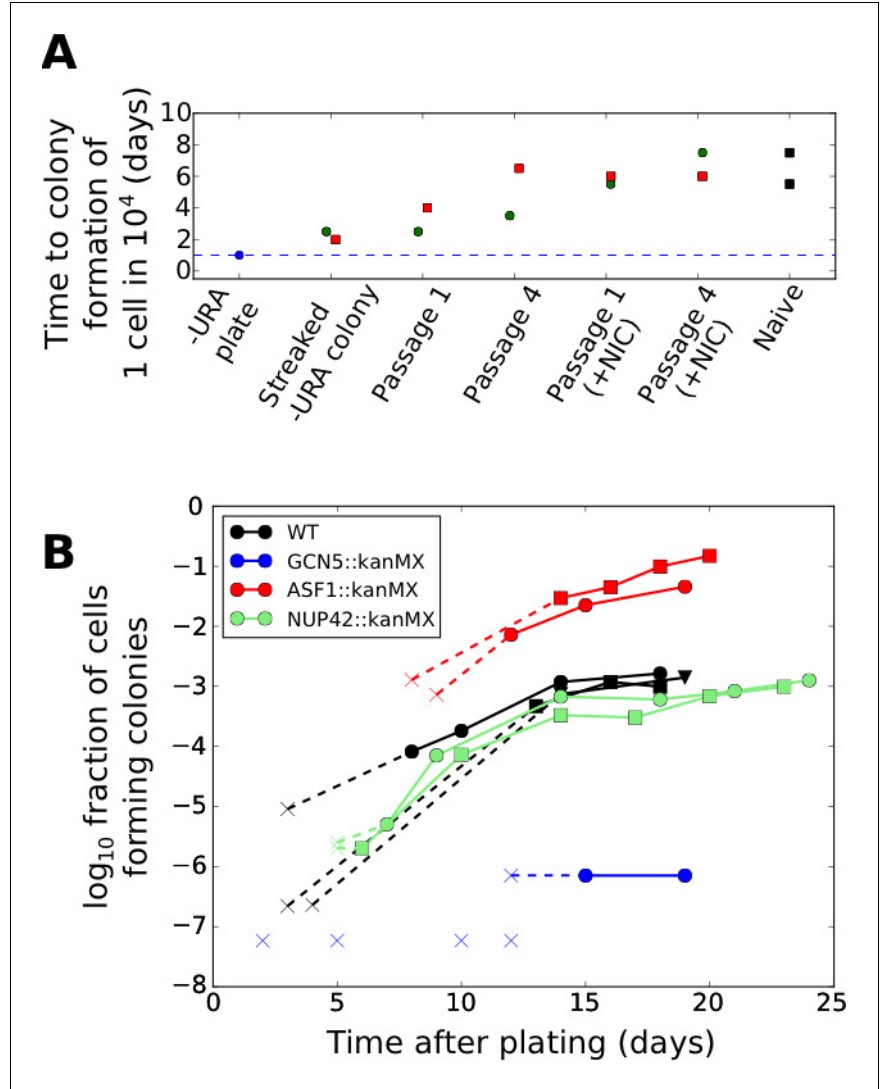

**Figure 8.** Effects of genetic and chemical perturbations on the efficacy of fitness-directed stochastic tuning and its epigenetic reversion. (A) Time courses of recovery back to the naïve state for tuned synprom-URA3-mRuby/P$_{ADH1}$-DHFR-GFP cells grown in either SC+glu or SC+glu with 25 mM nicotinamide added (+NIC). Extremes are shown for the colony formation times of cells never exposed to –ura conditions (Naïve) and for single colonies isolated after streaking out cells from ura-/6AU15 plates onto SC+glu (Streaked –URA colony). Colors of points indicate a single lineage beginning from a single streaked out colony picked at the first SC+glu plate stage. The cells were then repeatedly passaged in liquid SC+glu media and assessed for colony formation rates on ura-/6AU15 plates on subsequent days, as detailed in *Figure 8—figure supplement 1*. (B) Colony formation rates on ura-/6AU15 plates in the presence of various genetic perturbations, assessed by colony counts from platings of selected dilutions of cells. An 'x' followed by a dashed line indicates no observed colonies and is shown at the threshold of detection from the experiment. All mutations are in a synprom-URA3-mRuby/leu2Δ0 background.

DOI: https://doi.org/10.7554/eLife.31867.015

The following figure supplements are available for figure 8:

**Figure supplement 1.** Recovery of cells taken from ura-/6AU plates toward a naïve state.
DOI: https://doi.org/10.7554/eLife.31867.016

**Figure supplement 2.** Survival of cells in –ura media in the presence of genetic perturbations.
DOI: https://doi.org/10.7554/eLife.31867.017

even after a single round of growth on an SC+glu plate, and subsequently becoming stronger with additional SC+glu passages.

To conclusively exclude mutational mechanisms, we performed untargeted whole-genome resequencing of a total of eight isolates with synprom-driven URA3-mRuby (four colonies from 6AU15 plates and four separate biological replicates taken after the onset of growth in 6AU5 liquid media; see Materials and Methods for details). For each case, we scanned the region within 25 kb of the LEU2 locus (where the URA3 cassettes were integrated) for mutations, since control of URA3 expression was shown in these cells to operate locally in cis (*Figure 4* and *Figure 4—figure supplement 2*). The results are summarized in *Supplementary file 4*: Of the eight isolates, five show no mutations within 25 kb of the URA3-mRuby insertion, two show SNPs of unknown fitness contribution in a minority of the population, and one shows a duplication of the URA3-mRuby cassette (based on the presence of a read density that is twice the level observed elsewhere for the same chromosome). These data clearly indicate that the origin of growth-supporting URA3 expression levels in these cells cannot be reliant on a mutational mechanism, as only one of the eight cases – that with the URA3 duplication – shows a mutation at high enough levels in the population to explain the onset of growth (mutations present in less than half of the population must have arisen after one or more cells in the population had already tuned and began growing, and thus by definition could not be responsible for the initial onset of the growing state). The phenotypes caused by the sequence variants observed in populations C2 and L4 are not immediately obvious, but even if they are beneficial, their presence in a minority of cells excludes the possibility that they were responsible for the onset of tuning. Note that it should not be surprising (and, indeed, would be expected) that beneficial mutations might arise in a population once it had begun expanding in a new environment due to stochastic tuning. Our findings are consistent with a non-genetically heritable basis for the observed tuning in seven out of eight of the cases examined, as in all other growing lines, mutations near the URA3 gene were either non-existent or present only in a minority of the population.

## Excluding growth-selection on the basis of pre-existing variation in URA3 expression level

A formal possibility for colony formation in a subset of the population is that growth occurs solely on the basis of pre-existing URA3 levels in cells prior to being exposed to uracil deprivation. Microscopic observations of starving cells (*Figure 3E*) argue against this possibility, as a substantial lag passes before any cells begin sustained growth. Also, colony formation continues over the course of many days (*Figure 3B–D*), demonstrating that even cells that were non-growing for a substantial time period after exposure to URA- stress can eventually grow under this condition. Nevertheless, to conclusively discount the possibility of pre-existing URA3 levels determining tuning, we sorted populations of cells on the basis of initial URA3 expression, isolated those with the highest mRuby levels (the top 0.5–1%, well outside of the main distribution of the population) and plated them. These experiments clearly showed that the ability to form colonies on ura-/6AU plates is not restricted to cells with initially high URA3-mRuby expression (*Supplementary file 5*), as the highly fluorescent cells do not form colonies on ura-/6AU plates at rates substantially higher than unsorted cells, and certainly not at a sufficiently higher rate to fully explain the observed colony formation rates. These data argue against the possibility that growth occurs only in cells that, by chance, already have high levels of URA3 expression at the time of plating (although such cells may have some slight advantage, given the nature of their initial state).

## Stochastic tuning is affected by genetic perturbations to chromatin modification machinery

The proposed fitness-directed tuning mechanism relies on the capacity of local chromatin to maintain a memory of recent changes in transcription, and to modulate the transcription rate based on the fitness consequences of those changes, as conveyed by the proposed central metabolic integrator of health/fitness. We hypothesized that chromatin modification machinery may be intimately involved in these processes.

To probe the mechanistic basis of stochastic tuning, we focused on perturbations to histone acetylation/deacetylation (deletions of GCN5, SIN3, HST3, HST4), and chromatin remodeling (deletions of ASF1, ISW2, SWR1, UBP8), all of which provide potential pathways for coupling feedback from

the cell's physiological state to allele-specific modulation of chromatin and transcription (See *Table 1* for details). We selected these targets because of their association with genes showing particularly high levels of noise (and thus, more likely to be driven by tuning) in single-cell proteomic analysis (*Newman et al., 2006*). In our screening, homozygous replacements of HST3, HST4, SWR1, ISW2, and UBP8 with a kanMX cassette showed little effect on colony formation rates on ura-/6AU plates, and SIN3::kanMX/SIN3::kanMX strains showed severely compromised cell survival under growth-arrested conditions; all were excluded from further analysis. On the other hand, we found that genetic perturbations to the histone acetylation machinery through deletion of the key histone ace-tyltransferase GCN5 essentially abolished tuning, whereas deletion of the histone chaperone ASF1, in contrast, increased tuning rate by more than an order of magnitude (*Figure 8B*). At the same time, we show that the observed tuning process does not rely on transcriptional memory mecha-nisms grounded in chromatin localization, given the lack of effect of a NUP42 deletion (*Figure 8B*; *cf.* (*Guan et al., 2012*)).

## Variations in colony formation rate are not a result of changes in viability

In interpreting our data on the effects of genetic perturbations on tuning (*Figure 8B*), it was crucial to consider the possibility that cells may lose viability at variable rates under different conditions, which could contribute to the observed differences in colony formation rates. We thus performed experiments to measure the rate of cell death in the presence of uracil starvation and compared the results with the different colony formation rates observed. As shown in *Figure 8—figure supple-ment 2*, the effects of a mutation on survival and tuning rates are not significantly correlated. For example, deletion of GCN5 resulted in the nearly complete loss of stochastic tuning, deletion of NUP42 had no effect, and deletion of ASF1 substantially enhanced tuning, yet none of these muta-tions shows a change in survival rates during incubation in uracil-free media compared with wild type cells sufficient to explain the observed change in colony formation rate (*Figure 8—figure supple-ment 2*). Even for the poorest surviving strain, GCN5::kanMX/GCN5::kanMX, colony formation rates after ten days are 100−1000 times lower than wild type cells even though survival rates are lower only by a factor of ten.

## Chemical perturbation of histone deacetylases inhibits the maintenance of the tuned state

Given the apparent importance of chromatin modifications in fitness-directed tuning, we also tested the effects of nicotinamide treatment (which inhibits the sirtuin class of histone deacetylases, or HDACs (*Bitterman et al., 2002*)) on reversion of the tuned cells back to a naïve state. As shown in *Figure 8A*, we found that chemical inhibition of sirtuin HDACs by nicotinamide treatment substan-tially accelerated the decay of a tuned population to the naïve state, further highlighting the impor-tance of histone modification in stochastic tuning. Combined with the data on knockout strains described above, our results suggest a central role for chromatin modifications in the establishment

**Table 1.** Summary of genetic perturbations tested for effects on tuning rates.

| Perturbation | Direct effect | Effect on tuning |
| --- | --- | --- |
| GCN5::kanMX | Deletion of histone acetyltransferase subunit (acts in ADA, SAGA, SLIK/SALSA complexes) | Inhibits |
| SWR1::kanMX | Deletion of H2AZ exchange factor | No effect |
| UBP8::kanMX | Deletion of SAGA complex de-ubiquitinase | No effect |
| SIN3::kanMX | Deletion of Rpd3S/L histone deacetylase components | No effect |
| HST3::kanMX | Deletion of Sir2-family histone deacetylase | No effect |
| HST4::kanMX | Deletion of Sir2-family histone deacetylase | No effect |
| ISW2::kanMX | Deletion of DNA translocase involved in chromatin remodeling | No effect |
| ASF1::kanMX | Deletion of nucleosome assembly factor | Accelerates |
| NUP42::kanMX | Deletion of nuclear pore complex component known to be involved in transcriptional memory | No effect |

DOI: https://doi.org/10.7554/eLife.31867.018

and maintenance of the tuning process, although the molecular details cannot be discerned from these data alone.

## A biologically feasible implementation of stochastic tuning

The abstract model introduced in *Figures 1–2* demonstrates the potential utility of fitness-directed stochastic tuning to establish adaptive gene expression states without directly sensing the external environment. In order to substantiate the biological feasibility of stochastic tuning, we implemented its critical components in a plausible simulation incorporating generic features of chromatin modification and the information flow of the Central Dogma of Molecular Biology. We therefore designed and simulated a dynamical model tracking transcription rates, transcript levels, protein levels, and histone modifications in a single cell, with parameter distributions sampled from experimental data (*Figure 9A*; see Methods for details). We incorporated the possibility of adding or removing chromatin marks that can alter the transcription rates of the associated genes. Our model incorporates two classes of marks: tuning marks (T), which link cellular fitness to transcriptional output by having mark addition rates that are a function of the recent direction of change in global fitness and current number of such marks at each promoter; and stabilizing marks (S), which are added at a rate dependent on the number of tuning marks at each promoter (*Figure 9B*). At any time, the transcriptional output of the promoter is a function of the density of both tuning marks and stabilizing marks. As such, the tuning marks provide a critical connection between changes in global fitness and transcription rates, whereas the more slowly changing stabilizing marks capture the average transcriptional output over longer timescales, enabling a more stable optimization trajectory. Both T and S chromatin marks come in two varieties: positive (activating) and negative (repressive).

Our aim was to develop a generic simulation consistent with our general knowledge of coupling between chromatin modification and transcription (*Li et al., 2007*; *Rando and Winston, 2012*; *Zhou and Zhou, 2011*; *Mitra et al., 2006*). As such, the tuning and stabilizing marks described here need not correspond to any specific chemical moiety or be attributed to any particular histone modification enzyme. Modulation of enzyme activity by global fitness could be due to some as yet unknown signaling pathway or, alternatively, be dependent on known metabolic substrates or cofactors, such as acetyl-CoA and NAD+ (*Lin et al., 2000*; *Thaminy et al., 2007*; *Tanner et al., 1999*).

As shown in *Figure 9C*, the detailed model is capable of stochastic tuning of a single gene which strongly impacts the fitness of the cell (as would be the case for URA3 in our experimental setup). For most randomly generated gene-level parameters, stochastic tuning results in substantially higher fitness compared to when cells undergo random fluctuations in transcription levels or when transcription is fixed at a rate appropriate for a different environment, and in most cases, tuning is able to consistently achieve near-optimal expression levels. The model is robust to variations in both the sampled biological parameters (*Figure 9C*) and the parameters of the model itself (*Figure 9D*) and can locate an optimal expression level regardless of the ratio between the initial and target protein levels (*Figure 9E*). These results demonstrate that a generic, biologically feasible implementation of fitness-directed stochastic tuning can in fact function even in the presence of the multiple layers of noise and temporal delays acting between transcription rates (at which tuning occurs) and protein levels (which dictate fitness). Note that we do not expect to find conditions where stochastic tuning is the primary mechanism of gene expression modulation for every gene in the genome, even for novel or extreme environments. Rather, we expect that the cells' hard-wired transcriptional regulatory logic exerts the primary role in the transcriptional reprogramming of the majority of genes in the genome. For its part, we expect that stochastic tuning plays the dominant role in modulating the expression of few genes/pathways that represent critical bottlenecks for fitness (for example, induction of a drug efflux pump, or repression of an enzyme that activates a pro-drug chemotherapeutic agent).

## Discussion

We have described a mechanism of adaptation through fitness-directed optimization of gene expression. In numerical simulations, the proposed framework has the remarkable capacity to simultaneously tune the expression of thousands of genes, enabling optimization of fitness without directly sensing environmental parameters. The demonstration that a phenomenon consistent with fitness-directed stochastic tuning operates in *S. cerevisiae* has important implications for the

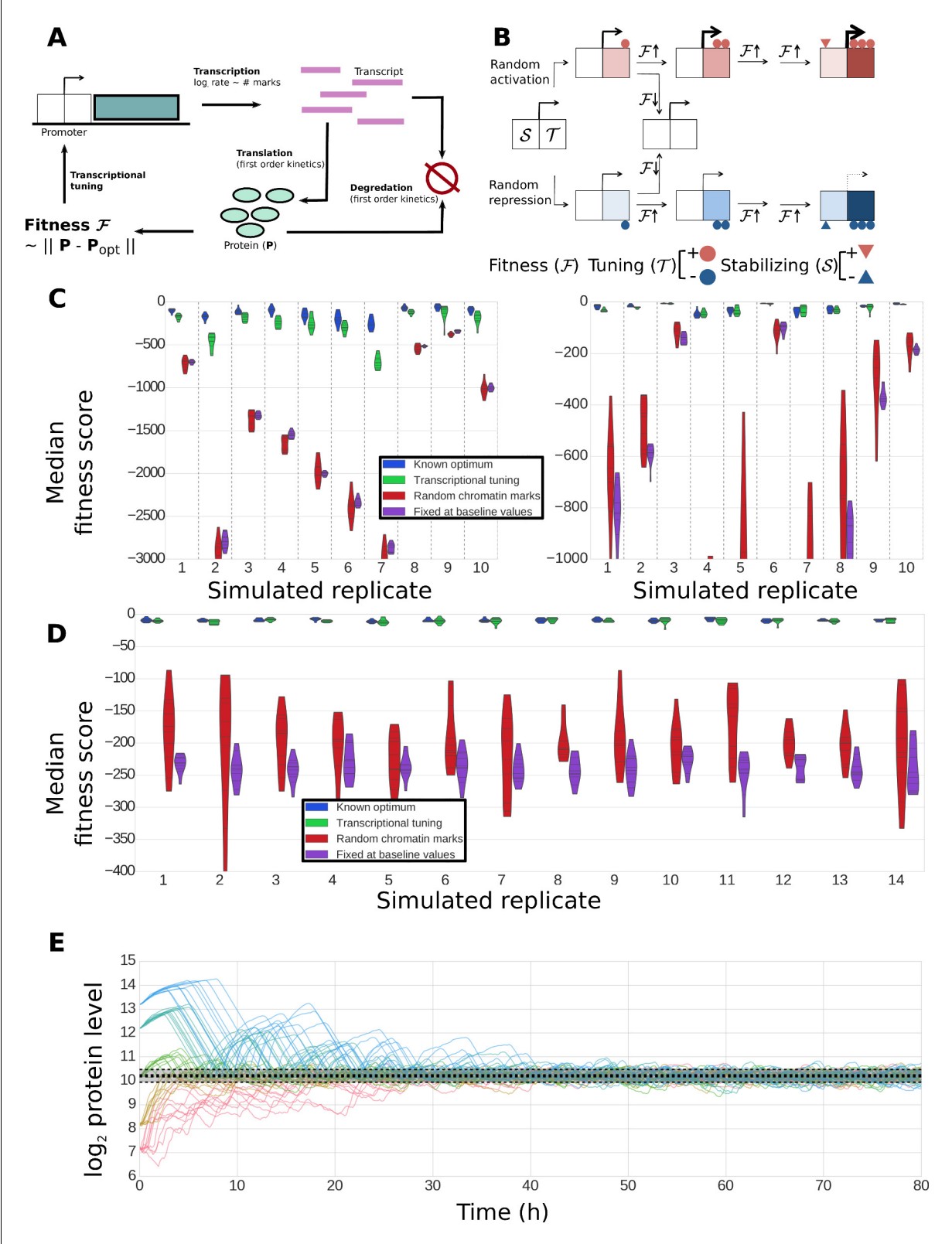

**Figure 9.** Construction and performance of a biologically feasible model for fitness-directed stochastic tuning. (**A**) Schematic of processes modeled in the simulation. Transcripts are produced at a rate dependent upon the state of chromatin marks at each promoter; each transcript has a fixed, gene-dependent probability of being translated at each timestep (producing a protein), and may also be degraded (again, with a gene-dependent probability). Similarly, each copy of a protein may be degraded at each timestep with a protein-dependent probability. The fitness of the system is

*Figure 9 continued on next page*

*Figure 9 continued*

calculated as the Euclidean distance between the current profile of protein counts present in the cell from a target optimum. Chromatin marks may be added or removed at each promoter at each step, as shown in panel B). (B) Logic underlying changes in tuning and stabilizing mark counts at each step. Tuning marks (T) may be added or removed at each step based on the recent history of changes in fitness, and whether each promoter currently has a net positive (activating) or negative (repressive) T count. Stabilizing marks (S) provide longer term integration by adding activating or repressive marks over time in response to the state of the tuning marks. Thus, if an unmodified promoter undergoes random addition of a positive tuning mark (top path), and that addition proves favorable, it will undergo further addition of positive T marks. If fitness continues to increase, stabilizing marks (S) will be added to stabilize its higher activity. Similar logic holds for the random addition of negative tuning marks (bottom path). In both cases, if the random T-mark perturbation proves unfavorable, the promoter will be modified in the opposite direction, in this case returning it back to its original unmodified state. (C) Distributions of fitness scores for a one-gene system obtained in twenty simulations using different randomly sampled biological parameters (e.g., transcript stabilities, translation rates, etc.) – these different parameter sets are the 'simulated replicates' referred to on the x axis. The median scores over the last quarter of the simulation are shown for 10 independent tuning trajectories (differing in their random number seeds). Each simulation proceeded for 300,000 steps (83.3 hr of simulated time). Different colors indicate varying methods used to control transcription rates (as shown in the legend): 'Known optimum' refers to a case where transcription rates are kept fixed at their predefined target values, 'Stochastic tuning' is the full model described in the Methods section, 'Random chromatin marks' is equivalent to the tuning model except that the direction of T chromatin mark addition is random instead of fitness directed, and 'Fixed at baseline values' shows the case where transcription rates are fixed at their initial values (intended to correspond to the environment that the cells were in prior to the onset of stress exposure). Dashed vertical lines group simulations performed with identical parameters. On the left axis we show ten sets of simulations where the target transcription rate was eight-fold higher than the starting rate, and on the right axis simulations where the target transcription rate was eight-fold lower than the starting rate. (D) Robustness of tuning against changing model parameters. Violin plots are defined as in panel C), but in this case show the distributions of fitness scores observed under variations of the model parameters (e.g., magnitude of individual tuning and stabilizing marks) for a single, randomly chosen set of gene-specific parameters. Plotted are the median fitness scores over the last quarter of each simulation, using either our central 'baseline' parameters for all model parameters (leftmost replicate; see *Supplementary file 8*), or twofold changes (up or down) of each editable parameter in our model. (E) Tuning performance of a single gene matching a wide range of biological challenges. For a fixed set of biological parameters (see Materials and Methods), we performed 10 simulations each where the initial transcription rates were off from the target rate by a factor of $2^3$, $2^2$, 0, $2^{-2}$, and $2^{-3}$, running in order from blue to red. A strong dashed black line shows the median obtained from the last quarter of a long (3 million step) simulation with transcription rates fixed at their optimal values; the shaded region shows the extent of a region encompassing 95% of the timepoints observed in that window. Regardless of initial conditions, the protein level approaches the optimal value and then stably oscillates around it, with amplitudes similar to those observed in the control simulation with target transcription rates.

DOI: https://doi.org/10.7554/eLife.31867.019

adaptation of eukaryotic microbes to novel or extreme environments where their genetically encoded regulatory networks become inadequate. However, we speculate that stochastic tuning operates in parallel with conventional regulation even in frequently encountered environments. Indeed, hard-coded sensory and regulatory networks are unlikely to have the encoding capacity to optimally respond to every conceivable subtle change in the environment—even within the native habitat. We therefore favor a model in which dedicated regulatory networks quickly move the system to a state reasonably well matched to a given condition, and stochastic tuning subsequently optimizes expression to achieve a more precisely adapted state for every individual encounter.

The ability to discover optimal gene expression states through a stochastic fitness-directed search may have provided significant advantage to early eukaryotic microbes. Microorganisms have evolved stochastic search strategies in other contexts. Indeed, the proposed stochastic tuning mechanism is reminiscent of the biased random walk phenomenon in bacterial chemotaxis, where stochastic transitions in the rotation of the flagellar motor are biased towards the direction that increases chemoattractant signaling over time (*Macnab and Koshland, 1972*). Detailed molecular mechanisms of chemotaxis have been revealed over the course of the last few decades, demonstrating the versatility of molecular processes in implementing rather complex computations (reviewed in (*Sourjik and Wingreen, 2012*)). Although our main focus here has been on establishing the phenomenology of fitness-directed stochastic tuning, we have already identified some critical components. In particular, histone acetylation/deacetylation (via GCN5 and sirtuins) seem to play a critical role, as deletion of GCN5 almost entirely abolished tuning. This is consistent with the high degree of intrinsic noise exhibited by the genes that are regulated by the SAGA complex, in which GCN5 is the catalytic subunit (*Newman et al., 2006*). Previous work has shown that increased transcriptional noise is beneficial for adaptation to acute environmental stress (*Blake et al., 2006*). Interestingly, however, early work demonstrated that deletion of GCN5 further increases expression noise in the context of the PHO5 promoter (*Raser and O'Shea, 2004*).

Taken together, these data suggest that stochastic tuning is not driven by noise alone; rather we support a model in which the proper integration of noise, transcriptional memory, chromatin modification, and cellular-health feedback work together to implement a directed search mechanism to drive the expression level of individual genes to levels that maximize the overall health of the cell. Indeed, histone modification is tightly coupled with gene expression. Co-transcriptional histone modification can store recent memory of transcriptional activity (*Li et al., 2007*; *Rando and Winston, 2012*) and histone modification can, in turn, affect transcription rate (*Stasevich et al., 2014*). There has been a longstanding debate on the functional significance of this reciprocal coupling. Our model and results help to unify these phenomena and support their functional relevance as requisite components of a stochastic tuning-based cellular adaptation framework.

We note that our experimental setup for demonstrating stochastic tuning has superficial similarities to a series of experiments performed in *S. cerevisiae* by the Braun lab, in which they sought to determine whether glucose-driven repression of the GAL1 promoter could be overcome to allow expression of a HIS3 construct in glucose-containing media (*Stern et al., 2007*; *Stolovicki et al., 2006*). While the authors observed consistent emergence of growth in a large fraction of cells that they initially noted could be attributed to either genetic or epigenetic mechanisms (*Stolovicki et al., 2006*), subsequent analysis has shown that in that experimental system, genetic mutations are the primary mechanism of adaptation, possibly driven by hypermutability of the genes involved in the response of interest (*David et al., 2010*; *Moore et al., 2014*; *David et al., 2013*). These mutational mechanisms stand in clear contrast to the rapidly reverting epigenetic stochastic tuning observed in our experiments.

In addition to perception of environmental parameters, cells also possess a variety of hard-wired homeostatic mechanisms sensing and responding to internal parameters, optimizing resource allocation in response to parameters such as growth rate (*Klumpp et al., 2009*; *Klumpp and Hwa, 2014*; *Brauer et al., 2008*; *Barenholz et al., 2016*; *Keren et al., 2013*) and metabolite/nutrient pools (*Potrykus et al., 2011*; *Broach, 2012*). However, while these mechanisms allow cells to sense their internal state, they still reflect specific evolved responses to alter resource allocation and gene expression in a *predefined* way in response to stress, standing in contrast with the ability of stochastic tuning to conduct a search and discover arbitrary gene expression states that are adaptive under extreme and unfamiliar environments.

The widely varying tuning rates for different promoters (*Figure 3B–C* and *Figure 3—figure supplement 1*) clearly indicate that sequence features can influence tuning efficacy. By design, all but one promoter driving URA3 in our experiments contained a TATA box, which has been linked to high intrinsic noise (*Newman et al., 2006*), condition-specific expression variability (*Tirosh et al., 2006*) and reliance on chromatin-mediated regulation (*Tirosh et al., 2008*; *Basehoar et al., 2004*). Indeed, replacement of the (TATA-containing) $P_{SAM3}$ derived sequence in synprom with a similarly generated sequence from the TATA-free $P_{ARF1}$ promoter substantially reduced tuning rates under the conditions tested (*Figure 3—figure supplement 1*). We also note that when we performed experiments similar to those described above with the repressed natural promoter $P_{GAL1}$, we observed dramatically lower rates of colony formation (less than 1 in $10^7$), and those colonies that did form appeared to be non-reverting genetic mutants (data not shown). Exploring the full importance of transcriptional noise for tuning efficiency, as well as that of other features such as propensity for nucleosome positioning, will be important in future work.

Fitness-directed stochastic tuning requires feedback of the global state of health to every promoter in the genome. The dependence of many histone modification enzymes on metabolic intermediates and cofactors (e.g., NAD+ for the sirtuin family of histone deacetylases (*Lin et al., 2000*; *Thaminy et al., 2007*); SAM for histone methyltransferases (*Luka et al., 2009*), and acetyl-CoA for histone acetyltransferases (*Tanner et al., 1999*)) provides support for potential direct feedback of global fitness-related parameters to the epigenome (*Katada et al., 2012*; *Kurdistani, 2014*), and indeed we showed that chemical manipulation of sirtuin activity had substantial effects on retention of epigenetic memory. These enzymes may very well serve as distinct channels of health-related information utilized by stochastic tuning. In this regard, chromatin itself may function as a global health integrator, with histone modifications and their effect on gene expression being highly contingent on the current trajectory of cellular fitness. Alternatively, cells may utilize a single global health integrator (such as the mTOR system) as hypothesized in our idealized model. The mTOR pathway integrates diverse parameters of internal health including energy, nutrient availability, and cellular

stresses (*González and Hall, 2017*). Intriguingly, the mTOR pathway has recently been shown to regulate histone acetylation states through a variety of mechanisms (*Chen et al., 2012*; *Workman et al., 2016*)

Fitness-directed stochastic tuning has important implications for gene regulation. Beyond a potentially widespread mechanism of cellular adaptation, stochastic tuning brings together seemingly unrelated phenomena under a unifying conceptual framework. These are areas of study at the frontier of genetics and biochemistry, including stochastic gene expression, transcriptional memory, and metabolic modulation of epigenetic states. Stochastic tuning may have initially evolved as a mechanism for adaptation of single-cell eukaryotes to extreme environments. However, once available, it may have found additional utility as a versatile mechanism for controlling and fine-tuning gene expression in the context of physiological and developmental processes in metazoans. This is consistent with the evolutionary arc of an ancient set of molecular mechanisms that now serve as key mediators of differentiation (*Álvarez-Errico et al., 2015*; *Ziller et al., 2015*; *Meissner, 2010*). Exploring this possibility represents an important area for future research. Optimization of cellular health through the fitness-directed stochastic tuning mechanism may also play an important role in allowing cancer cells to survive and thrive in a variety of microenvironments unfamiliar to their evolved regulatory networks, and in the face of extreme challenges imposed by chemotherapy and radiation. Indeed, stochastic tuning may underlie the epigenetically mediated metastatic potential and chemotherapy resistance observed in a variety of cancer types (*Wu and Roberts, 2013*; *Perez-Plasencia and Duenas-Gonzalez, 2006*; *Lv et al., 2016*; *Li et al., 2015*; *Borley and Brown, 2015*; *Bonito et al., 2016*; *Shaffer et al., 2017*). Our observations support the existence of a fitness-directed tuning process that operates at the level of transcription. However, in principle, tuning could also occur at any point along the hierarchy of gene expression where noise, memory, and feedback of global fitness can drive the activity of gene products towards levels that optimize the overall health of the cell. In particular, searching for evidence of tuning at the level of translation would be an important focus for future research.

# Materials and methods

**Key resources table**

| Reagent type (species) or resource | Designation | Source or reference | Identifiers | Additional information |
|---|---|---|---|---|
| gene (*Saccharomyces cerevisiae*) | URA3 | NA | YEL021W | |
| gene (*Entacmaea quadricolor*) | mRuby | DOI: 10.1371/journal.pone.0004391 | | |
| gene (*Aequorea victoria*) | GFP | DOI: 10.1038/nbt1172 | | Codon optimized for *S. cereivisiae*; sequence available as **Supplementary file 3** |
| genetic reagent (*S. cerevisiae*) | P$_{HSP12}$ | NA | | Promoter region upstream of YFL014W |
| genetic reagent (*S. cerevisiae*) | P$_{ADH1}$ | NA | | Promoter region upstream of YOL086C |
| genetic reagent (*S. cerevisiae*) | P$_{RGI1}$ | NA | | Promoter region upstream of YER067W |
| genetic reagent (S. cerevisiae) | synprom | This paper | | Synthetic promoter sequence. See Supplementary Material for complete sequence, and methods for details of construction |
| genetic reagent (S. cerevisiae) | GCN5::kanMX | PMID: 10436161 | | Knockout cassette obtained from the yeast knockout collection strain |
| genetic reagent (S. cerevisiae) | ASF1::kanMX | PMID: 10436161 | | Knockout cassette obtained from the yeast knockout collection strain |
| genetic reagent (S. cerevisiae) | NUP42::kanMX | PMID: 10436161 | | Knockout cassette obtained from the yeast knockout collection strain |
| strain background (*S. cerevisiae*) | BY4743 | PMID: 9483801 | | |
| chemical compound, drug | 6-azauracil | ACROS Organics | Product code 153970050 | Stock solution 10 mg/mL in 1 M ammonium hydroxide |

*Continued on next page*

*Continued*

| Reagent type (species) or resource | Designation | Source or reference | Identifiers | Additional information |
|---|---|---|---|---|
| chemical compound, drug | Nicotinamide | Sigma | Product number N0636 | Stock solution 1 M in water; filter sterilized |
| software, algorithm | tuning_simple | This paper | | Octave implementation provided as *Source Code 2* |
| software, algorithm | tuning | This paper | | Python implementation provided as *Source Code 3* |

## Media and strains

For routine growth of strains, we used YPD broth (10 g/L yeast extract, 20 g/L peptone, 20 g/L dextrose) or YPD agar plates (YPD broth +20 g/L Bacto agar). We used standard recipes based on SC +glucose (SC+glu) (*Kaiser et al., 1994*) for all physiological experiments. SC/loflo refers to SC made with low fluorescence yeast nitrogen base (US Biologicals). In the case of SC+glu, we used dropout supplement powders interchangeably from ForMedium (DSCK012) and US Biologicals (D9515), although they differ slightly in the concentrations of adenine and *para*-amino benzoic acid supplied. SC+glu derivatives lacking particular nutrients are specified as SC+glu-NUTRIENT; e.g., SC+glu-ura for SC+glu lacking uracil. We also refer to the commonly used mixture of SC+glu-ura with 6-azauracil added as ura-/6AU$i$, where $i$ is the final concentration of 6AU in microgram/mL. The agar for all plates used in physiological experiments was either Noble agar (Difco) or quadruple-washed Bacto agar. For the removal of the GAL-GIN11 cassette in counter-selections (see below), cells were plated on YPGA agar plates (10 g/L yeast extract, 20 g/L peptone, 20 g/L galactose, 20 g/L agar, 100 microgram/mL ampicillin). All growth was at 30°C; liquid phase growth included shaking at 200–220 rpm in an Innova 42 incubator (New Brunswick).

As diagrammed in *Figure 3A*, we constructed two classes of insertion cassettes. Each follows the pattern of having a promoter, a functional reporter protein fused to a fluorescent protein, and then ends with a CYC1 terminator. For URA3, the native sequence from *S. cerevisiae* was used, with the exception of one silent SNP and an A160S mutation that does not appear to alter enzyme function. The red fluorescent protein mRuby is described in (*Kredel et al., 2009*). For DHFR, we used murine DHFR from pSV2-dhfr (*Subramani et al., 1981*) with an L22R mutation making it methotrexate-resistant (*Simonsen and Levinson, 1983*). GFP refers in all cases to superfolder GFP (*Pédelacq et al., 2006*) codon-optimized for *S. cerevisiae* using web-based tools from IDT (Integrated DNA Technologies); see *Supplementary file 3* for the corresponding nucleotide sequence. In each case, the reporter and fluorescent protein were separated by a short A/G/S containing linker. All constructs were cloned in bacterial hosts using pBAD-derived plasmids; separate plasmids were constructed with each promoter of interest downstream of a region homologous to the upstream target site in the *S. cerevisiae* genome, and URA3-mRuby-cyc or DHFR-GFP-cyc upstream of a region homologous to the downstream target site in the *S. cerevisiae* genome. All constructs were chromosomally integrated at the leu2Δ0 locus of our yeast strains. Double-stranded DNA for transformation in yeast was then generated by first amplifying the promoter and reporter constructs separately, using primers yielding 20–40 bp overlaps; we then used crossover PCR to generate the complete construct of interest and subsequent amplification to generate a sufficient quantity for transformation. All PCR used for strain construction was performed using Q5 high fidelity polymerase (NEB); routine PCRs for strain validation were instead performed using OneTaq or Taq polymerase (NEB).

Promoters for ADH1, HSP12, and RGI1 were cloned from our wild type strain (BY4743 or its haploid progenitors BY4741/BY4742) and included the entire region from 1700 to 1800 bp upstream of the start codon to the base immediately prior to the start codon. The ADH1 promoter was selected as a classic constitutive promoter (*DeMarini et al., 2001*); HSP12 and RGI1 were chosen as they show high variance in expression between conditions (*Tirosh et al., 2009*; *Tirosh et al., 2006*), a characteristic expected to be favorable for stochastic tuning. Synprom was designed in two stages: the bulk of the DNA is a 600 bp random sequence generated using a Markov model to match the trinucleotide frequencies present across all natural *S. cerevisiae* promoters. To this sequence we appended the 200 bp immediately prior to the start codon of SAM3, to provide native transcription and translation start sites. The resulting sequence was then modified to remove all recognizable

binding sites for yeast transcription factors (TFs) as follows: we used the set of position weight matrices and match thresholds in ScerTF (*Spivak and Stormo, 2012*) to identify all recognizable TF binding sites in the promoter, and randomized the sequences of only those regions and their immediate surroundings until no recognizable TF binding sites remained. The resulting perturbed sequence is given as *Supplementary file 1*. The required sequences were synthesized as gBlocks from Integrated DNA Technologies and combined via Gibson assembly (*Lartigue et al., 2009*).

All yeast strains were derived from BY4741 or BY4742 (*Brachmann et al., 1998*), which includes a complete deletion of the URA3 ORF (BY4741: Mat **a** his3Δ1 leu2Δ0 met15Δ0 ura3Δ0; BY4742: Mat α his3Δ1 leu2Δ0 lys2Δ0 ura3Δ0). Insertions of URA3 or DHFR fusion proteins were always at the leu2Δ0 locus unless otherwise noted. To facilitate consistent insertion, we replaced the leu2Δ0 allele of BY4741/BY4742 with a LEU2-GAL-GIN11 cassette (*Akada et al., 2002*), which allows growth in leucine-free media but inhibits growth in the presence of galactose. We note that at least in our copy of the BY474x strains, the leu2Δ0 deletion runs only from ChrIII:84799—ChrIII:93305, rather than extending to position 93576 as annotated. Nevertheless, the deletion is sufficient to remove the entire leu2 open reading frame.

Strains containing the fusion proteins were constructed by transforming the LEU2-GAL-GIN11 containing cells with appropriate double-stranded oligos (see above) and selection on YPGA plates, allowing replacement of the LEU2-GAL-GIN11 cassette with the desired insert. Insertions were confirmed by PCR product sizing. Diploid strains were derived by mating one BY4741-derived (mat **a**) strain with one BY4742-derived (mat α), and subsequently plating on SC+glu-lys-met or SC+glu-lys-met-cys. All transformations were carried out using the LiAc-PEG-ssDNA method (*Gietz and Woods, 2002*).

Knockout strains were generated by beginning from appropriate haploids containing either a leu2::promoter-URA3 or leu2::promoter-DHFR construct or simply leu2Δ0, amplifying an appropriate kanMX knockout cassette from the corresponding strain in the *S. cerevisiae* gene deletion collection (*Giaever et al., 2002*), and selecting on YPD+G418 plates. We confirmed the presence of kanMX at the appropriate site and absence of the native gene by PCR. Diploid knockout strains containing appropriate deletions and a URA3-mRuby insertion at leu2Δ0 were generated by mating these haploids as noted above.

## Colony formation assays

Experiments showing colony formation rates over time all follow a common formula. Cells were grown overnight in SC+glu media, and then in the morning back-diluted 1:200 into fresh, prewarmed SC+glu. The cells were grown for four to five hours at 30℃ with shaking and then pelleted, washed once with 25 mL deionized (DI) water, pelleted, washed with 1 mL water, pelleted, and resuspended in 1 mL water. Specified dilutions were made in DI water from this final cell suspension.

Cells were then either plated on full plates at pre-chosen dilutions (100 microliters of an appropriate cell suspension), or a dilution series was spotted onto appropriate agar plates (10 microliters per spot). Plates were imaged and counted every 1–2 days for the duration of the experiment (lasting between a few days and weeks, depending on the experiment in question). Plates were wrapped in parafilm after ~3 days to minimize drying. Plating was performed identically on SC+glu plates (to establish the number of cells being plated) and plates containing one or more test conditions (e.g., ura-/6AU).

Cells were counted either directly from the plates or from stored digital images. Direct plate counts were done manually for all visible colonies; for those counted from saved images, we imposed a minimum size threshold of 0.2 mm in diameter (rounding up to the nearest pixel). Times for counts were rounded to the number of days since plating.

## Death rate assays

To determine the survival rates of cells undergoing uracil starvation in the presence of various other perturbations, we measured the death rates of cells lacking any copy of URA3 in SC-ura+glu media. Cells were pregrown and washed as described above for plating assays, but then resuspended in liquid SC-ura+glu media and incubated at 30℃. Aliquots were regularly removed and spotted on SC+glu plates to determine the number of viable colonies. Survival rates are for leu2Δ0 homozygotes

(the original BY4743 diploid, possibly with a homozygous deletion of a specified gene) with no available copy of URA3.

## Flow cytometry

Cells were analyzed by flow cytometry on an LSR Fortessa (Becton Dickinson) at the Columbia University Microbiology and Immunology Flow Cytometry Core Facility or University of Michigan Flow Cytometry Core. Cells to be used in these experiments were initially prepared and washed following the same pregrowth procedure as given above for colony formation assays, except that growth was in low fluorescence SC/loflo media instead of SC. A flask containing 25 mL of prewarmed media (generally ura-/6AU5 made from an SC-ura/loflo base) was then inoculated with 200 microliters of the cell suspension, and cells were grown with shaking at 30°C. Subsequent data acquisition varied depending on the experiment to be performed.

For the long time courses shown in *Figure 4* and its supplement, for an initial timepoint, 200 microliters of the washed cell suspension were combined with 500 microliters of 2x PBS/E (1x PBS with 10 mM EDTA added), 290 microliters DI water, and 10 microliters of flow cytometry counting beads (Invitrogen CountBright beads). At subsequent timepoints, snapshots were taken by combining 490 microliters of the growing cells, 10 microliters counting beads, and 500 microliters 2x PBS/E. In either case, cells were run on the Fortessa, with signals recorded for forward and side scatter, mRuby (using the Texas Red laser/filter set), and GFP (using the FITC laser/filter set).

Data were analyzed using the flowCore and flowViz modules of R (*Ellis et al., 2006*; *Ellis et al., 2009*). Beads and cells were first identified based on their forward scatter and side scatter (FSC/SSC) values (using permissive gates that capture the vast majority of each population) and fluorescence (beads were required to show very high fluorescence). For each growth phase (exponential in SC+glu, starving in ura-/6AU, growing in ura-/6AU), we obtained empirical autofluorescence corrections by analyzing populations in a similar growth state lacking the fluorescent tag on URA3. Guided by exploratory analysis, we fit a linear model for starving cells predicting mRuby and GFP autofluorescence as a function of the observed forward and side scatter, and used constant autofluorescence values characteristic of each of the two growing phases (obtained from cells with no fluorescent protein in a similar physiological state, either uracil-starved or undergoing stochastic tuning-driven growth). During analysis of liquid phase fluorescent populations (shown in *Figure 4* and its supplement), the predicted autofluorescence values were subtracted from the observed value; in these cases, an additional gate was applied to remove events with very low forward scatter values, which had a very high variance in fluorescence and were well below the size of the main population.

For the use of FACS followed by plating to test the colony formation rates of highly fluorescent cells, cells were prepared as described above, sorted using a BD FACSAria, and then subsequently plated in equal quantities on SC+glu and ura-/6AU15 plates.

For the short timescale tuning data shown in *Figure 5C*, the cells were grown for 3–4 hr side by side in SC/loflo + glu and –ura/loflo/6AU1 media, and then placed on ice and run directly on the flow cytometer. For each biological replicate (performed on different days), we grew leu2::synprom-URA3-mRuby/leu2::synprom-DHFR-GFP and nonfluorescent leu2::URA3/leu2Δ0 cells in parallel to allow direct comparison of the observed fluorescence levels. Analysis was performed separately for each biological replicate. We first normalized all fluorescence signals by the FSC-A signal raised to the power of 1.5, which we found empirically to be an effective correction removing most of the dependence of the fluorescence on cell size. Next, a mapping of FSC signals to expected autofluorescence on each channel was fitted using the R loess function (with default parameters), and the expected autofluorescence subtracted from the observed value for each cell to yield what we refer to as the blanked fluorescence. We then calculated and compared the changes in the median blanked fluorescence of the populations for the same cells grown in SC+glu vs. ura-/6AU1 media. Confidence intervals were calculated by bootstrapping with 200 bootstrap replicates.

## Whole genome sequencing

Cells for whole genome sequencing were taken directly from the growth condition of interest (ura-/6AU15 plate or ura-/6AU5 liquid media) and flash frozen in 15% glycerol or 1x TES (10 mM Tris, pH 7.5; 10 mM EDTA, 0.5% SDS). One reference sample grown under unselective conditions was taken for each starting strain to use as a baseline. Genomic DNA was isolated using a YeaStar Genomic

DNA kit (Zymo Research) according to the manufacturer's instructions. Samples were then barcoded and prepared for sequencing using a Nextera XT kit (Illumina, Inc.) and sequenced as part of a pooled library on a NextSeq (Illumina, Inc.).

Sequencing reads were clipped to remove adapters and commonly observed artifactual end sequences with cutadapt (*Martin, 2014*), and then further trimmed using Trimmomatic 0.30 (*Bolger et al., 2014*) to remove very low quality (<3) end bases, retain only the portion of the read with a quality score above 15 in a four base sliding average window, and remove reads less than 10 bp long. Surviving trimmed reads were then aligned to the reference genome using Bowtie 2.1 (*Langmead et al., 2009*); the reference genome was constructed from the *S. cerevisiae* S288c genome (GenBank BK006934 – BK006949), deleting the URA3 ORF and inserting the sequence for the appropriate URA3 and DHFR constructs in separate copies of chromosome III at the LEU2 locus. Read data used in this analysis are available from the Short Read Archive under accession SRP117724.

After alignment, mutational calls and read depths were obtained using the mpileup and depth modules of samtools 0.1.18 (*Li et al., 2009*), respectively. Reads for called variants within 25 kb of the insertion site were examined manually and compared to the sequenced parental strain; validated variants are listed in *Supplementary file 4*.

## RNA isolation

RNA was isolated using an adaptation of the hot acid phenol method (*Collart and Oliviero, 2001*). Cells for RNA isolation were grown under appropriate conditions (either in liquid phase or on agar plates), and then snap-frozen in 1x TES (10 mM Tris, pH 7.5; 10 mM EDTA; 0.5% SDS) and stored below −70°C. Snapshots of 200 to 600 microliters were taken from growing liquid phase cultures, whereas from agar plates we harvested 1–20 colonies of <0.5 mm diameter taken from the same plate as each biological replicate. RNA was isolated by rapidly thawing the cell suspension and mixing 1:1 with a 5:1 acid phenol:chloroform solution, then incubating 60 min at 65°C with occasional vigorous vortexing. The solution was then chilled on ice for 5 min, and centrifuged 5 min at 16,000 x g at 4°C. The aqueous phase was mixed 1:1 with additional acid phenol:chloroform, chilled, and centrifuged as before. The aqueous phase was then mixed 1:1 with a 24:1 chloroform:isoamyl alcohol solution, and centrifuged 5 min at 4°C. The resulting aqueous phase was transferred to a fresh tube and combined with 1/10 vol 3 M sodium acetate, 2 volumes of 1:1 ethanol:isopropanol, and 1/800—1/200 vol Glycoblue (Ambion), and then precipitated for at least 1 hr at −20°C and then at least 1 hr at −80°C. RNA was recovered by centrifuging 15 min at 16,000 x g at 4°C, washed with ice cold 75% ethanol, spun an additional 5 min, and then air-dried and resuspended in RNAse-free water. The samples were then further purified using a Zymo RNA clean and concentrator five according to the manufacturer's instructions, including an on-column DNase digestion.

## Quantitative RT-PCR

Total RNA was purified from cells in the desired growth condition using the hot acid-phenol procedure described above. cDNA pools were generated for each sample using random hexamer-primed reverse transcription with Protoscript II (New England Biolabs) following the manufacturer's instructions. cDNA pools were used directly in qPCR reactions without further purifications, assembling reactions using iTaq Universal SYBR Green Supermix (BioRad) following the manufacturer's instructions, in GeneMate PCR plates. Plates were sealed with Microseal 'B' adhesive film (BioRad) and run on a BioRad CFX96 detection system. $C_t$ values calculated by the instrument software were then exported for subsequent analysis. All isolated RNA was quantified on a Bioanalyzer (Agilent) and found to have an RIN >= 6.8.

For comparison of URA3 and DHFR expression, we calculated separate $\Delta C_t$ values for each qPCR run replicate by taking the median of all technical replicates from that run. Values plotted in *Figure 4—figure supplement 2* reflect $\Delta C_t$ data from 1 to 2 technical replicate wells on each of two to four separate, independently assembled runs; we plot the median of day-wise data points for each separate biological sample. Primer locations and sequences are given in *Supplementary file 7*. We performed a no-reverse transcriptase control reaction for each sample to ensure that DNA contamination did not contribute to the observed signal (data not shown).

qRT-PCR data were analyzed using a Bayesian hierarchical model treating the $\Delta C_t$ value between the URA3 and DHFR primers as follows:

$\Delta C_t(sample,day) \sim T(\mu_s(sample), \sigma_{rep}, \nu_{rep})$

$\mu_s(sample) \sim T(\mu_c(class), \sigma_c(class), \nu_{bio})$

Parameters not otherwise specified were assigned appropriate uninformative priors. Here 'sample' refers to a single biological sample and 'class' to a single growth condition. The key parameter of interest is $\mu_c$ for each class of cells under study, the overall average URA3:DHFR difference for cells grown under that condition. We fitted the model using JAGS (*Plummer, 2003*), and then report credible intervals and other inferences from the posterior distribution on $\mu_c$. Each of the $\Delta C_t(sample, day)$ values used the median across 1–2 technical replicates for each primer pair.

### Cell count data analysis

Data were analyzed using custom-written python and R scripts. Source code for the nontrivial analysis of flow cytometry data is provided as *Source code 1*. Uncertainties for cell counts (shown in plating and flow cytometry data) were calculated by treating each observed count as a Poisson random variable; using Bayesian inference with the Jeffreys prior (*Jeffreys, 1961*), the posterior distribution for the rate parameter I (the concentration of cells) is given in closed form by

$I \sim Gamma(0.5 + \sum_{i=0}^{n} i_n, n)$

Where $n$ is the number of observations and the $i_n$ are the observed counts. Error bars then indicate a central 95% credible interval for I given the observed data.

### Recovery experiments

Experiments to examine the reversion of tuned colonies toward a naïve state were performed as shown in *Figure 8—figure supplement 1*. Single colonies from a ura-/6AU15 plate were streaked out onto SC +glu and allowed to grow. From that plate, single colonies were again picked and underwent repeated passages in liquid media; each 'passage' refers to a 200-fold dilution, which is then allowed to grow for 48 hr (96 hr for the very first transfer). Cells were also taken for plating from the original ura-/6AU15 plate, the first SC +glu plate stage, and several subsequent time points during liquid culture. Cells taken from plates were immediately diluted in water and spotted on SC +glu and ura-/6AU15 to track colony formation rates; cells taken from liquid passages were streaked out on SC +glu plates prior to use in spottings, in order to obtain a consistent physiological state. Plots for 'naïve' cells refer to cells treated identically, except that they had initially been grown on SC +glu plates instead of ura-/6AU15 plates. Recovery was assessed based on the amount of time required for 1 in 10,000 cells spotted on the new ura-/6AU15 plate to form countable colonies (using linear interpolation of colony counts between observed data points); in the event that one dilution yielded no colonies passing our size threshold, but the next (10-fold more concentrated) spot gave an uncountable haze, we assigned a count of 1 to the more concentrated spot.

### Numerical simulations

The numerical simulations shown in *Figures 2* and *5* were performed by implementing the model described in the text using the Matlab programming language and simulated using Matlab (Mathworks, Inc.) or GNU Octave version 3.8.1 (*Eaton et al., 2009*), with qualitatively equivalent results obtained in either case. All simulations were performed using the same initial conditions (but different random seeds, for the sampling shown in *Figure 5*). Octave code implementing this model is provided as *Source code 2*.

The physiological tuning model employed for *Figure 9* and the accompanying text was implemented in python, and simulated using python 2.7.6, making heavy use of the numpy (*Svd et al., 2011*) and scipy (*Jones et al., 2001*) libraries, with data analysis and plotting using matplotlib (*Hunter, 2007*) and pandas (*McKinney, 2010*). The details of the physiological model itself are given below.

### Biologically feasible simulation of stochastic tuning

To provide a suitable mechanistic model for stochastic tuning, we developed a discrete-time model tracking the temporal evolution of transcription rates $r_{i,t}$ (continuous, changed in response to

random fluctuations and potentially tuning input), copy number of each transcript per cell $x_i$, and copy number of each protein per cell $p_i$, considered separately for each gene $i$.

Transcriptional regulation lies at the center of our consideration for fitness-directed tuning. In the physiological model, there is a time-dependent probability $r_{i,t}$ for a single transcript to be generated from gene $i$ at each timestep; the probabilities $r_{i,t}$ are updated in response to changing fitness as described below. In addition, each copy of the transcript present in the cell has a fixed probability $d_i$ of being degraded at each timestep. The net change at each timestep $t$ in the transcript level $x_i$ for each gene $i$ is thus given by

$$x_{i,t} \sim x_{i,t-1} - \text{binom}(x_{i,t-1}, d_i) + \text{bern}(r_{i,t-1})$$

Here binom/bern are binomial and Bernoulli random variables, respectively. Terms using binomial distributions allow a uniform probability for each present copy of a protein or transcript to be degraded or translated, whereas the Bernoulli term captures the probability of a transcript arising from each gene in a single timestep. We used a timestep of 1 s for all simulations described here.

Protein production in our physiological model arises from similar principles. At each timestep, each copy of a transcript from gene $i$ has a fixed gene-dependent probability $l_i$ of being translated to produce a single copy of the corresponding protein. In addition, each copy of that protein already present in the cell has a gene-dependent probability $e_i$ of being degraded. Thus, the net rate of change in the protein copy number $p_i$ at each time $t$ is governed by the equation

$$p_{i,t} \sim p_{i,t-1} - \text{binom}(p_{i,t-1}, e_i) + \text{binom}(x_{i,t-1}, l_i)$$

The fixed, gene-specific parameters $d_i$, $e_i$, and $l_i$ were drawn from distributions that are themselves fits to appropriate experimental data; we then modified the fitted parameters to yield distributions that are contained within the physiological distributions, while excluding the extreme ends of the available range. The parameters used for the physiological rate distributions are summarized below:

**Transcription rates** (used to initialize the transcription rate distribution, and separately to set the target transcription rate distribution): Transcripts per hour are gamma distributed with shape = 5 and rate = 2 (obtained by fitting data from (*Holstege et al., 1998*) and excluding extreme values)

**Transcript degradation rates** $d_i$: Half lives in minutes have a gamma distribution with shape = 12.0 and rate = 0.75 (obtained by fitting data from (*Holstege et al., 1998*) and then modifying to exclude extreme values).

**Protein degradation rates** $e_i$: Half lives in hours have a scaled t distribution with mean = 1, sigma = 0.382, and 80 degrees of freedom (fit based on data from (*Christiano et al., 2014*), but modified to exclude long half-lives, consistent with the induction of autophagy in stressed cells (*Cebollero and Reggiori, 2009*)).

**Protein synthesis rates** $l_i$: log2 synthesis rates per transcript have a scaled t distribution with mean=-5, sigma = 0.5, and 80 degrees of freedom (in units of s$^{-1}$); based on protein abundance data from (*Kulak et al., 2014*) combined with the other parameters defined above, and modified to exclude extreme values).

As described in the main text, our model permits two classes of 'marks' (representing histone modifications) that alter transcription rates: tuning marks (T), which change in level on the basis of recent changes in fitness and the current tuning mark state at each gene, and stabilizing marks (S), which change in abundance based on the tuning mark levels at each promoter. The number of each mark type at each promoter may be positive or negative, reflecting the possibility of distinct activating (+) or repressing (-) chromatin modifications.

The rate of change in the tuning marks proceeds according to the following principles. At each timestep, marks may be added or removed on the basis of recent changes in fitness; each mark may decay with a fixed probability; and marks may be added or removed in an undirected manner due to random drift. Referring to the number of tuning marks at a particular gene $i$ as $\mu_i$, the change in tuning marks at each timestep due to the tuning contribution alone is given by

$$\Delta\mu_{i,tuning} \sim \text{sgn}(\Delta F_t) * \text{sgn}(\mu_i) * \text{randint}(1,5) * \text{bern}(p_{tunestep})$$

Here sgn($x$) is one if $x$ is positive, $-1$ if $x$ is negative, and 0 if $x$ is zero. $\Delta F$ indicates the difference in mean fitness between the previous $n_{window}$ steps and the nonoverlapping block of $n_{window}$ steps before that; thus, sgn($\Delta F_t$) will be positive if the cells are becoming healthier, and negative if the cells are becoming less healthy. The fitness itself, $F_t$, is calculated as the Euclidean distance between the observed vector of protein levels $\boldsymbol{p_t}$ at a particular timestep, and the median observed in the last quarter of a long (10 times the normal simulation length) trajectory where all transcription rates are

fixed at their target values (note that oscillation still occurs, even in this case of known-correct transcription rates, due to the inherent randomness in transcript and protein production and degradation). In the context of our model, $\Delta F$ represents the direction of change in global cellular health, and $p_{tunestep}$ indicates the probability that tuning marks will be added/removed at a particular timestep. The combination of signs of the change in fitness ($\Delta F$) and marks ($\mu_i$) ensures that if the fitness is increasing and a given promoter has a positive number of tuning marks, the number of tuning marks at that promoter will increase further, whereas if the fitness was decreasing, the number of tuning marks will be decreased. The inverse directions apply for promoters with negative levels of T marks. Note that for control simulations where the effects of tuning are removed, the sign of the fitness-dependent term above is instead taken to be random.

The removal and random drift of tuning marks are governed by the equations

$\Delta\mu_{i,removal} \sim -1 * sgn(\mu_i) * binom(\mu_i, p_{decay})$

and

$\Delta\mu_{i,random} \sim (1-2*bern(0.5)) * bern(p_{random})$ respectively. The first equation here indicates that each individual mark may be removed with probability $p_{decay}$ at each timestep, and in addition, the second equation dictates that each promoter may have a single mark of random sign added at each timestep, with probability $p_{random}$. The overall equation for the change in tuning marks at promoter $i$ at each timestep is thus given by the sum of the terms above:

$\Delta\mu_{i,t} \sim \Delta F_t * sgn(\mu_{i,t-1}) * randint(1,5) * bern(p_{tunestep}) - sgn(\mu_i) * binom(\mu_i, p_{decay}) + (1-2*bern(0.5)) * bern(p_{random})$

The stabilizing marks (S), in contrast, do not vary directly in response to fitness, but rather, at each timestep may be added or removed from each promoter depending on its current state of T marks (see *Figure 9B*): if the promoter has a high transcription rate due to high T levels, the net S count will be increased (with a probability at each timestep proportional to the current magnitude of the T level), and if the promoter has low T levels, the net S count is decreased. The effect of the stabilizing marks is to slowly shift the baseline transcription rate of genes over time. The change in number of S marks $\nu_i$ at gene $i$ at each timestep is given by:

$\Delta\nu_{i,t} \sim sgn(\mu_{i,t-1}) * bern(abs(\mu_{i,t-1}) * p_{s\_mark} / \mu_{max})$

Here $p_{s\_mark}$ is a probability of changing S marks at each time step, and $\mu_{max}$ the maximum number of T marks allowed at a given promoter, whether positive (activating) or negative (repressive).

Every gene in the model is taken to have a baseline transcription rate, $r_{i,0}$, drawn from the physiological distributions defined above. The time-dependent instantaneous transcription rate of a given gene, $r_{i,t}$, is then calculated from the number of tuning marks ($\mu_i$) and stabilizing marks ($\nu_i$). The effects of tuning and stabilizing marks in the model are multiplicative, such that the transcription rate $r_i$ at gene $i$ with $\mu_i$ tuning marks and $\nu_i$ stabilizing marks is given by

$r_{i,t} = r_{i,0} * \alpha * exp(\beta)$; where $\alpha = 2*((\mu_{i,t} / \mu_{max})+1)$ and $\beta = m_S * \nu_{i,t}$

Here $m_S$ represents the magnitude of the effects of a single S mark, and the number of T marks is constrained to the interval $[-\mu_{max}, \mu_{max}]$. The various fixed model parameters (e.g., $m_S$, $p_{decay}$, etc.) were chosen to be physiologically plausible while supporting tuning. The values of these parameters used in *Figure 9C and E* are taken as a baseline and shown in *Supplementary file 8*; note, however, that as shown in *Figure 9D*, the performance of the model is robust to changes in those parameters.

A python implementation of the model, along with sample inputs corresponding to the simulations described here, are included as *Source code 3*.

## Fluorescence tracking of sorted populations

In order to measure the mixing times under different stress conditions, synprom-URA3-mRuby/synprom-DHFR-GFP cells were grown overnight in SC +glu media. The next morning, the cells were back-diluted 1:100 into fresh, prewarmed low fluorescence SC +glu or ura-/6AU10. The cells in ura-/6AU10 media were kept in a 30°C incubated shaker for 24 hr before sorting, whereas the cells in the complete media were sorted after four hours of growth at 30°C. The cells were sorted based on their mRuby fluorescence level into three populations of the top 20%, bottom 20%, and the complete distribution (mock-sorted) of cells. In order to minimize the effects of both autofluorescence and size-fluorescence correlations, the cells (including those in the mock-sorted population) were tightly gated on FSC-A levels. The sorted cells were kept on ice until they were spun down and transferred to pre-warmed media identical to that in which they had previously been incubated (that is, cells from complete media to complete media and cells from ura-/6AU to fresh ura-/6AU). The cells were

incubated at 30℃ thereafter. A sample of each population was analyzed using flow cytometry at different time intervals, with T = 0 being the time that the fresh media was added to the samples. The last time point for the cells in SC +glu media was 630 min, and for the ura-/6AU cells was 6660 min.

We calculated the distribution of mRuby fluorescence values for each sample at each time point by smoothing the observed values using a kernel density estimator. We then measured the pairwise mRuby fluorescence distribution overlap of the top 20%, bottom 20% and the complete distribution at each time point for each growth condition. The distribution overlap was calculated by numerically integrating the area under the (normalized) kernel density distribution estimates of both populations being compared.

An increasing overlap relative to t = 0 signifies the amount that the two populations have moved towards each other, and therefore the higher the overlap, the more mixed the two populations have become. Therefore, we calculated $f(t) = \frac{(\max(x) - x(t))}{x_{t=0}}$, where x is the overlap between the two distributions and max(x) is the maximum observed overlap. $f$(t) can be modeled as an exponential decay process according to:

$$f(t) = ae^{-\frac{t}{\tau}}$$

where $\tau$ provides a timescale for the mixing time (in particular, $\tau$ ln(2) is the half-life of the decay process). We used nonlinear curve fitting in Matlab to estimate the values of the parameters in the above equation for cells grown under each of the physiological conditions described above and report the estimated half-lives to give insight into the mixing times active in the populations studied here.

## Fluorescence microscopy time courses on immobilized cells

The images shown and analyzed in *Figure 6*, *Figure 7*, and panel A of *Figure 7—figure supplement 2* were obtained on a Zeiss Axio Observer Z1, using a 40x objective lens. $P_{HSP12}$-URA3-mRuby/$P_{ADH1}$-DHFR-GFP cells were grown overnight in SC +glu liquid media, and then back-diluted 100x into SC/loflo + glu media and grown four additional hours with shaking at 30℃. Cells were spun down, and then incubated in ura-/loflo/6AU5 liquid for 12–13 hr. The cells were then pipetted onto the prepared slides. In order to prepare slides, we added 200 µL of ura-/loflo/6AU5 media containing 1% agar to each well of a two-well slide. Using a 22 µm coverslip, the surface of the media in the wells containing the solid media was flattened. After adding the cells on to the wells, we allowed extra media to be absorbed and then added a cover slip on top. The cells were imaged on DIC, GFP, brightfield, and mRuby channels; snapshots were taken once every 30 min for approximately 24 hr.

The additional imaging time series analyzed in panels B-D of *Figure 7—figure supplement 2* were obtained for $P_{HSP12}$-URA3-mRuby/$P_{ADH1}$-DHFR-GFP cells immobilized to thin-bottomed growth chambers and grown in ura-/6AU5 media. To prepare the slides, cells were grown overnight in SC +glu liquid media, and then back-diluted 100x into SC/loflo + glu media and grown four additional hours with shaking at 30℃. During that incubation, a coverslip/incubation chamber (Nunc) was treated for five minutes with poly-D-lysine solution (MPI Biomedical), washed three times with sterile deionized water, and then allowed to dry.

After the pregrowth period, cells were diluted 10x into additional prewarmed SC/loflo + glu, and then pipetted onto the poly-D-lysine treated cover slip and allowed to settle for 30 min at room temperature. The media was removed, and non-adherent cells were washed away with two 1 mL rinses of sterile deionized water. The cells were then covered with 2 mL of ura-/loflo/6AU5 media, and then placed in a preheated microscopy incubation chamber (OKO) at 30℃ and 90% relative humidity. Cells were imaged on DIC, GFP, and mRuby channels; snapshots taken once every 30 min for 24 hr on a Nikon Eclipse Ti microscope using a 20x objective.

For comparative visualization purposes (*Figure 7A–B*), the DIC or brightfield channel of each image was rescaled using the ImageMagick 'normalize' operator, and the fluorescence channels were normalized by subtracting the minimum pixel value within a given field of view, and then subjecting the remaining data to a median filter over a 5 × 5 pixel window. The fluorescence channels were then stacked on the DIC or brightfield to generate the images shown. Un-normalized data were used for all quantitative analysis.

For the quantitative analysis in *Figure 7C–E* and *Figure 7—figure supplement 2*, segmentation and lineage tracking were performed manually to identify cell division events and define cell interiors at the plotted timepoints. The fluorescence of each cell for each channel was then taken to be the average value of all pixels within the defined cell interior, with the mode value of all pixels in a defined window around the cell subtracted as background. For the purpose of classifying cells based on their division state, a cell was classified as 'dividing' if it gave rise to a daughter cell before the next analyzed snapshot. Timepoints prior to three hours were excluded from quantitative analysis of dividing vs. nondividing cells for the populations pregrown in SC/loflo + glu, as a large fraction of cells in all of our microscopy experiments did undergo a single division before arresting, likely using residual nutrients from their previous growth in complete media.

## Acknowledgements

We thank members of the Tavazoie laboratory and Sohail Tavazoie for helpful discussions and feedback on the manuscript, and Panos Oikonomou for crucial input on the use of sorted cells to track mixing times. We are also grateful to Kaushik Ragunathan for additional helpful discussion and access to microscopy equipment. PLF was supported by a K99 Award (K99/R00-GM097033) from NIGMS. JY was supported by the NIH MSTP program at Columbia University Medical School. ST was supported by grants from NIH/NIAID (R01-AI077562) and the NIH Director's Pioneer Award (DP1-ES022578).

## Additional information

### Funding

| Funder | Grant reference number | Author |
|---|---|---|
| National Institute of General Medical Sciences | K99 (GM097033-01A1) | Lydia Freddolino |
| National Institutes of Health | MSTP | Jamie Yang |
| NIH Office of the Director | 8DP1ES022578 | Saeed Tavazoie |
| National Institute of Allergy and Infectious Diseases | R01-AI077562 | Saeed Tavazoie |

The funders had no role in study design, data collection and interpretation, or the decision to submit the work for publication.

### Author contributions

Lydia Freddolino, Conceptualization, Data curation, Software, Formal analysis, Validation, Investigation, Visualization, Methodology, Writing—original draft, Writing—review and editing; Jamie Yang, Data curation, Validation, Methodology, Writing—review and editing; Amir Momen-Roknabadi, Formal analysis, Validation, Investigation, Visualization, Methodology, Writing—original draft, Writing—review and editing; Saeed Tavazoie, Conceptualization, Software, Formal analysis, Supervision, Funding acquisition, Investigation, Visualization, Methodology, Writing—original draft, Project administration, Writing—review and editing

### Author ORCIDs

Lydia Freddolino (iD) http://orcid.org/0000-0002-5821-4226
Saeed Tavazoie (iD) http://orcid.org/0000-0003-2183-4162

### Decision letter and Author response

Decision letter https://doi.org/10.7554/eLife.31867.051
Author response https://doi.org/10.7554/eLife.31867.052

# Additional files

## Supplementary files

• Source code 1. Custom R and python code for analysis of flow cytometry data.
DOI: https://doi.org/10.7554/eLife.31867.020

• Source code 2. Octave implementation of the simple stochastic tuning model used in *Figures 1*, *2* and *5*. See the accompanying README file for documentation.
DOI: https://doi.org/10.7554/eLife.31867.021

• Source code 3. Python implementation of the physiological stochastic tuning model used in *Figure 9*. See the accompanying README file for documentation.
DOI: https://doi.org/10.7554/eLife.31867.022

• Supplementary file 1. Complete sequence of synprom5-sam3, the promoter referred to as 'synprom' in the main text. Pseudorandomly generated sequence is shown on the first line, and SAM3-derived sequence on the second. The sequence has been perturbed to remove all recognizable transcription factor binding sites, as described in Materials and Methods.
DOI: https://doi.org/10.7554/eLife.31867.023

• Supplementary file 2. Sequences of other synthetic promoter components referenced in the text. Synthetic promoters consist of the combination of one pseudorandom 'synprom' sequence with either the SAM3 or ARF1 promoter proximal regions. All sequences have been perturbed to remove all recognizable transcription factor binding sites, as described in Experimental Procedures.
DOI: https://doi.org/10.7554/eLife.31867.024

• Supplementary file 3. Fitted parameters for distribution overlap half-lives from Figure S4. Shown are fitted values for the half-lives plus (in parentheses) the extent of a 95% confidence interval based on the model fit. All half-lives are given in minutes.
DOI: https://doi.org/10.7554/eLife.31867.025

• Supplementary file 4. Results of resequencing of tuned colonies and planktonic populations in the 25 kb vicinity of the URA3 and DHFR insertions. Numbers in parenthesis after mutation calls indicate the approximate fraction of the population containing the mutant allele. 'ID' is simply an identifier used to refer to each sample in the text.
DOI: https://doi.org/10.7554/eLife.31867.026

• Supplementary file 5. Colony counts for the extreme most highly fluorescent cells (top 0.5–1%) isolated from populations in which URA3-mRuby is driven by the specified promoter. Equal volumes of the sorted cells were plated in parallel on SC+glu and ura-/6AU15 plates, and then counted after 2–3 days (SC+glu) or 19–20 days (6AU). 'Baseline' refers to the fraction of cells expected to form colonies on 6AU15 plates in 19–20 days in unsorted populations (c.f. *Figures 3–4* of the main text).
DOI: https://doi.org/10.7554/eLife.31867.027

• Supplementary file 6. Codon optimized sequence of superfolder GFP used in all GFP constructs. Note that no start codon is included, as the construct is intended to be part of a fusion protein.
DOI: https://doi.org/10.7554/eLife.31867.028

• Supplementary file 7. Primer design for quantitative PCR experiments. End locations are given relative to the start codon of the gene in question.
DOI: https://doi.org/10.7554/eLife.31867.029

• Supplementary file 8. Baseline model parameters for the physiological tuning simulations described in *Figure 9*.
DOI: https://doi.org/10.7554/eLife.31867.030

• Transparent reporting form
DOI: https://doi.org/10.7554/eLife.31867.031

## Major datasets

The following datasets were generated:

| Author(s) | Year | Dataset title | Dataset URL | Database, license, and accessibility information |
|---|---|---|---|---|
| Freddolino L, Yang J, Tavazoie S | 2017 | Saccharomyces cerevisiae genome sequencing after transcriptional tuning | https://trace.ddbj.nig.ac.jp/DRASearch/study?acc=SRP117724 | Publicly available at the DNA Data Bank of Japan (accession no. SRA608529) |
| Freddolino L, Yang J, Tavazoie S | 2017 | NGS data from Stochastic tuning of gene expression enables cellular adaptation in the absence of pre-existing regulatory circuitry | https://www.ncbi.nlm.nih.gov/sra/?term=SAMN07652631 | Publicly available at the NCBI Sequence Read Archive (accession no: SAMN07652631) |
| Freddolino L, Yang J, Tavazoie S | 2017 | NGS data from Stochastic tuning of gene expression enables cellular adaptation in the absence of pre-existing regulatory circuitry | https://www.ncbi.nlm.nih.gov/sra/?term=SAMN07652632 | Publicly available at the NCBI Sequence Read Archive (accession no: SAMN07652632) |
| Freddolino L, Yang J, Tavazoie S | 2017 | NGS data from Stochastic tuning of gene expression enables cellular adaptation in the absence of pre-existing regulatory circuitry | https://www.ncbi.nlm.nih.gov/sra/?term=SAMN07652633 | Publicly available at the NCBI Sequence Read Archive (accession no: SAMN07652633) |
| Freddolino L, Yang J, Tavazoie S | 2017 | NGS data from Stochastic tuning of gene expression enables cellular adaptation in the absence of pre-existing regulatory circuitry | https://www.ncbi.nlm.nih.gov/sra/?term=SAMN07652634 | Publicly available at the NCBI Sequence Read Archive (accession no: SAMN07652634) |
| Freddolino L, Yang J, Tavazoie S | 2017 | NGS data from Stochastic tuning of gene expression enables cellular adaptation in the absence of pre-existing regulatory circuitry | https://www.ncbi.nlm.nih.gov/sra/?term=SAMN07652635 | Publicly available at the NCBI Sequence Read Archive (accession no: SAMN07652635) |
| Freddolino L, Yang J, Tavazoie S | 2017 | NGS data from Stochastic tuning of gene expression enables cellular adaptation in the absence of pre-existing regulatory circuitry | https://www.ncbi.nlm.nih.gov/sra/?term=SAMN07652636 | Publicly available at the NCBI Sequence Read Archive (accession no: SAMN07652636) |
| Freddolino L, Yang J, Tavazoie S | 2017 | NGS data from Stochastic tuning of gene expression enables cellular adaptation in the absence of pre-existing regulatory circuitry | https://www.ncbi.nlm.nih.gov/sra/?term=SAMN07652637 | Publicly available at the NCBI Sequence Read Archive (accession no: SAMN07652637) |
| Freddolino L, Yang J, Tavazoie S | 2017 | NGS data from Stochastic tuning of gene expression enables cellular adaptation in the absence of pre-existing regulatory circuitry | https://www.ncbi.nlm.nih.gov/sra/?term=SAMN07652638 | Publicly available at the NCBI Sequence Read Archive (accession no: SAMN07652638) |

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
