## [Decision Letter]

Thank you for submitting your article "Cellular adaptation through fitness-directed transcriptional tuning" for consideration by *eLife*. Your article has been reviewed by two peer reviewers, and the evaluation has been overseen by a Reviewing Editor and Naama Barkai as the Senior Editor. The reviewers have opted to remain anonymous.

The reviewers have discussed the reviews with one another and the Reviewing Editor has drafted this decision to help you prepare a revised submission.

Summary:

This paper presents a new striking theory on gene expression. It suggests that at the absence of an evolved cellular program to regulate gene expression according to the environmental conditions cells might resort to stochasticity to govern transcription of relevant genes. The paper suggests an intriguing feedback loop: stochastic noise allows certain cells to express the right gene to the right level (and other cells to the "wrong" level), these cells that, due to chance, expressed the protein to the right level get a signal to keep inducing the gene that improved their fitness. The presumed feedback allows cells to "know", somehow, that they should keep inducing that gene that conferred the high fitness or keep reducing genes if reduction improved their fitness; cells are also assumed to know to reverse changes in expression (up or down) if change if previous step was maladaptive.

Essential revisions:

The theory is intriguing, bold and novel. However, there are additional experiments that are essential to establish its use.

We find that experimentally addressing concerns 1and 2 of reviewer 2 is essential. The second two comments are optional – they will improve the study, but we leave it for you to decide.

Reviewer #1:

This study investigates an additional mechanism to traditional transcriptional regulation through the use of a randomized synthetic promoter controlling the expression of a bottleneck gene. This work provides substantial evidence supporting their hypothesis of allele-specific stochastic tuning driven by histone modification, and addresses several competing hypotheses, such as evolutionary pressure and global transcriptional changes. In total, we would recommend acceptance of this manuscript.

Reviewer #2:

This paper presents a new striking theory on gene expression. It suggests that at the absence of an evolved cellular program to regulate gene expression according to the environmental conditions cells might resort to stochasticity to govern transcription of relevant genes. The paper suggests an intriguing feedback loop: stochastic noise allows certain cells to express the right gene to the right level (and other cells to the "wrong" level), these cells that, due to chance, expressed the protein to the right level get a signal to keep inducing the gene that improved their fitness. The presumed feedback allows cells to "know", somehow, that they should keep inducing that gene that conferred the high fitness or keep reducing genes if reduction improved their fitness; cells are also assumed to know to reverse changes in expression (up or down) if change if previous step was maladaptive.

To test the notion, the authors built promoter-gene constructs in yeast in which they place a needed gene under a new promoter context. They starve the cell for the metabolite that is produced by the gene, forcing the cells to find new ways to express the needed gene. They use diverse unrelated promoters (noisy or synthetic) and they measure recovery from starvation under each. Despite being a novel challenge for the yeast, they find ways to overcome it within a few days. The authors claim that their observed recovery depends on promoter activity and noise, and they exclude mutations, and pre-challenge diversity as a source of adaptation. They strikingly show that the dynamics is dependent on several epigenetic regulatory proteins thus suggesting that histone modifications are mediating this stochastic tuning.

The theory is intriguing, bold and novel. The result with the chromatin modification proteins is striking. Though not complete the new theory is certainly exciting and should be brought to the attention of the community. Thus, publication in *eLife* is certainly supported in principle.

1) The most striking aspect of the presented theory is that there exists a postulated feedback from the fitness into the expression of the gene that determines the fitness. If true, this is ground breaking. But I'm still missing a direct experimental demonstration of this central claim. One way to show that could have been based on time lapse microscopy: follow cells as they express the fitness-affecting gene and measure simultaneously their growth rate and expression of that gene. A fitness-to-expression feedback could have been detected here if cells that happen to induce the right gene to the right amount were found to be more fit (say if cell doubling was measured and tracked from single cells) AND if the fitness-affecting gene was shown to further increase its expression level (or further decrease its expression as predicted from 𝛥𝐸! = 𝑘 ∙𝑠𝑔𝑛(𝛥𝐹t ∙ 𝛥𝐸t-1) + 𝜂) in those fit cells. A good negative control would have been to look in parallel at another unrelated gene whose expression does not affect fitness.

2) Time scales: a central question regarding this theory is on compatibility of the time scales involved. Noisy behavior of a promoter usually has "memory" (or "mixing times) in the order of a cell cycle, i.e. a cell that expressed highly a protein due to a stochastic fluctuation would "remember" that fluctuation for only one cell cycle and then decay back to the population average (such mixing times are typically observed, but perhaps not here). I think the current theory is based on the premise that a stochastic fluctuation, if beneficial, would be sustained for days due to the positive feedback from fitness (it took the cells here 5 days to show recovery). I'm not sure if time scales are compatible. Maybe a direct measurement of mixing time, along with the simulation, can clarify this point.

3) The presented theory is based on the notion that noisy promoters would generate the phenotypic diversity, from which adaptation takes place. I would have thus expected the authors to place the URA3 gene under the control of both high- and low-noise promoters for a comparison. Instead the two natural promoters are said to be chosen based on their conferring high noise. Aren't we missing a low noise promoter for comparison? The synthetic promoter might confer that property, but (i) is it conferring low noise? (a noise measurement would have been helpful for all promoters); (ii) even if it does, wouldn't it be better if they used a natural promoter that's characterized by low noise? Perhaps the replacement of the (TATA-containing) PSAM3 with TATA-free PARF1 is very helpful in that respect, but I'm not sure they showed that this TATA-free alternative is indeed noise reduced.

4) Towards excluding potential effect of mutations, the authors sequenced the strains around the relevant cis region. Could it be that the duplication detected at the URA3-mRuby cassette is a genetic change that confers advantage?

---

## [Author Response]

Essential revisions:Reviewer #2:This paper presents a new striking theory on gene expression. It suggests that at the absence of an evolved cellular program to regulate gene expression according to the environmental conditions cells might resort to stochasticity to govern transcription of relevant genes. The paper suggests an intriguing feedback loop: stochastic noise allows certain cells to express the right gene to the right level (and other cells to the "wrong" level), these cells that, due to chance, expressed the protein to the right level get a signal to keep inducing the gene that improved their fitness. The presumed feedback allows cells to "know", somehow, that they should keep inducing that gene that conferred the high fitness or keep reducing genes if reduction improved their fitness; cells are also assumed to know to reverse changes in expression (up or down) if change if previous step was maladaptive.To test the notion, the authors built promoter-gene constructs in yeast in which they place a needed gene under a new promoter context. They starve the cell for the metabolite that is produced by the gene, forcing the cells to find new ways to express the needed gene. They use diverse unrelated promoters (noisy or synthetic) and they measure recovery from starvation under each. Despite being a novel challenge for the yeast, they find ways to overcome it within a few days. The authors claim that their observed recovery depends on promoter activity and noise, and they exclude mutations, and pre-challenge diversity as a source of adaptation. They strikingly show that the dynamics is dependent on several epigenetic regulatory proteins thus suggesting that histone modifications are mediating this stochastic tuning.The theory is intriguing, bold and novel. The result with the chromatin modification proteins is striking. Though not complete the new theory is certainly exciting and should be brought to the attention of the community. Thus, publication in eLife is certainly supported in principle.We appreciate the reviewer’s reassuring feedback. The specific comments are addressed below.1 The most striking aspect of the presented theory is that there exists a postulated feedback from the fitness into the expression of the gene that determines the fitness. If true, this is ground breaking. But I'm still missing a direct experimental demonstration of this central claim. One way to show that could have been based on time lapse microscopy: follow cells as they express the fitness-affecting gene and measure simultaneously their growth rate and expression of that gene. A fitness-to-expression feedback could have been detected here if cells that happen to induce the right gene to the right amount were found to be more fit (say if cell doubling was measured and tracked from single cells) AND if the fitness-affecting gene was shown to further increase its expression level (or further decrease its expression as predicted from 𝛥𝐸! = 𝑘 ∙𝑠𝑔𝑛(𝛥𝐹t ∙ 𝛥𝐸t-1) + 𝜂) in those fit cells. A good negative control would have been to look in parallel at another unrelated gene whose expression does not affect fitness.

We thank the reviewer for this excellent suggestion, which allowed us to more closely inspect one of the key predictions of our model. As suggested by the reviewer, we have performed single-cell analysis of fluorescence microscopy time courses on cells that are either in the early stages of ongoing tuning (Figure 6, Figure 7 and on cells at the very onset of tuning (Figure 7—figure supplement 2). As suggested by the reviewer, we observe several central features consistent with our model: cells with higher levels of URA3-mRuby are more likely to divide and give rise to progeny with yet higher mRuby levels over the span of several generations. In addition, the ratio of URA3-mRuby (which is required for cellular health) relative to DHFR-GFP (which is dispensable) increases over subsequent divisions. We also note that another prediction of our model is that even before the onset of growth, once cells have achieved a sufficient level of URA3 such that it begins to improve their metabolic state, they should begin to specifically increase in URA3 expression. Precisely this phenomenon is observed in bulk populations in Figure 5C (where we see early and specific increases in URA3-mRuby in response to challenge with low levels of 6AU).

We believe that taken together, our microscopy and bulk data adequately demonstrate that URA3-mRuby levels in our system show lineage-specific and increasing levels of expression coinciding with the onset of growth, whereas the connection is weaker or nonexistent for the non-beneficial DHFR-GFP fusion protein in the same cells. These observations provide strong support for our proposed fitness feedback reinforcement mechanism. However, in the longer term it is critical that we identify the exact molecular components of this feedback and determine the detailed underlying mechanisms. This is the primary focus of ongoing work.

2) Time scales: a central question regarding this theory is on compatibility of the time scales involved. Noisy behavior of a promoter usually has "memory" (or "mixing times) in the order of a cell cycle, i.e. a cell that expressed highly a protein due to a stochastic fluctuation would "remember" that fluctuation for only one cell cycle and then decay back to the population average (such mixing times are typically observed, but perhaps not here). I think the current theory is based on the premise that a stochastic fluctuation, if beneficial, would be sustained for days due to the positive feedback from fitness (it took the cells here 5 days to show recovery). I'm not sure if time scales are compatible. Maybe a direct measurement of mixing time, along with the simulation, can clarify this point.

We thank the reviewer for the very insightful comment. We would like to begin here by pointing out that while the theory presented here would require a longer mixing time than a single cell cycle, mixing times of several days are not necessary. As indicated by the simulations in Figure 5A, and the stochastic nature of the tuning process in response to a severe stress (Figure 3 and Figure 4), the system’s behavior prior to the onset of tuning is dominated by a random walk rather than long-term memory, and sustained memory, over a long timescale, should only be observed when the expression of the growth-limiting gene fluctuates to a high enough level to initiate the proposed fitness feedback reinforcement mechanism. During exposure to a less severe stress, on the other hand, memory is only needed over time scales of a few hours (Figure 5C). Fluctuations on timescales of several hours are both expected based on, and tolerated by, our physiologically relevant model of tuning (see Figure 9E).

With those bounds established, we have now measured both the population-level mixing times of cells in our experimental setup under both normal (uracil replete) growth and uracil starvation, and the direct propagation of high-fluorescence states across several generations in growing microcolonies. As shown in Figure 7—figure supplement 1 and in Supplementary file 1, for batch cultures of cells growing in SC+glu we observe mixing times on the order of 1-2 hours (consistent with roughly one generation), whereas for cells exposed to uracil starvation stress those mixing times rise by nearly an order of magnitude, providing a longer cellular memory to support feedback between transient changes in fitness (which need not result in an immediate cell division event) and gene expression.

While those longer mixing times in starving cells enhance the feasibility of our tuning model, as the reviewer points out, ultimately the tuned state must propagate across several generations in order to enable growth. To determine whether such behavior is in fact observed, we tracked the inheritance of fluorescence levels during the formation of microcolonies in ura-/6AU5 media (Figure 6 and Figure 7), and observed that, as predicted by our model, high-fluorescence cells both show higher division rates and trans-generational inheritance of high-fluorescence states. We believe that the combination of these bulk and single-cell observations demonstrates the presence of sufficiently long ‘memory’/autocorrelation times in the presence of low-uracil challenge for our model to be feasible. We have added a lengthy discussion of this point to the new subsection “Tuning dynamics at the single cell level”.

3) The presented theory is based on the notion that noisy promoters would generate the phenotypic diversity, from which adaptation takes place. I would have thus expected the authors to place the URA3 gene under the control of both high- and low-noise promoters for a comparison. Instead the two natural promoters are said to be chosen based on their conferring high noise. Aren't we missing a low noise promoter for comparison? The synthetic promoter might confer that property, but (i) is it conferring low noise? (a noise measurement would have been helpful for all promoters); (ii) even if it does, wouldn't it be better if they used a natural promoter that's characterized by low noise? Perhaps the replacement of the (TATA-containing) PSAM3 with TATA-free PARF1 is very helpful in that respect, but I'm not sure they showed that this TATA-free alternative is indeed noise reduced.

We appreciate the reviewer’s suggestion that further elucidation of the noise levels inherent to the various promoters studied here would be helpful in some aspects of interpreting our findings. Unfortunately, as the synthetic promoters used here were by design quite low in expression, most of them in fact have URA3-mRuby expression levels so low that they cannot be reliably quantified under baseline growth conditions. Even for cases where both the noise and average expression level could be compared for different synthetic promoters, it is difficult to distinguish between the two effects on the limited set of promoters available.

Regarding the point of natural low-noise promoters, we did in fact perform similar experiments to those described here with the tightly repressed GAL1 promoter and TATA-free SPS4 promoter; both showed dramatically different behavior from the tuning observed for other promoters. Instead, colony formation rates for strains with URA3 driven by P_GAL1_ or P_SPS4_ were less than 1 in 10^6^(P_SPS4_) and less than 1 in 10^7^ (P_GAL1_), and the colonies that formed did not readily revert to the naive state (and thus may have arisen due to genetic mutations); these findings provide additional evidence for the importance of transcriptional noise in tuning.

We acknowledge the reviewer’s point that we cannot make firm statements regarding noise for the strains shown in Figure 3—figure supplement 1 in the absence of more systematic comparisons of the gene expression noise using different promoter-proximal regions with tuning efficacy. We have modified the associated text to make clear the limitations of our current state of knowledge, and identify possible future research along the same lines:

“The widely varying tuning rates for different promoters (Figure 3B,C and Figure 3—figure supplement 1) clearly indicate that sequence features can influence tuning efficacy. By design, all but one promoter driving URA3 in our experiments contained a TATA box, which has been linked to high intrinsic noise (Newman et al., 2006), condition-specific expression variability (Tirosh et al., 2006) and reliance on chromatin-mediated regulation (Tirosh et al., 2008; Basehoar, Zanton and Pugh, 2004). Indeed, replacement of the (TATA-containing) P_SAM3_ derived sequence in synprom with a similarly generated sequence from the TATA-free P_ARF1_ promoter substantially reduced tuning rates under the conditions tested (Figure 3—figure supplement 1). We also note that when we performed experiments similar to those described above with the repressed natural promoter P_GAL1_, we observed dramatically lower rates of colony formation (less than 1 in 10^7^), and those colonies that did form appeared to be non-reverting genetic mutants (data not shown). Exploring the full importance of transcriptional noise for tuning efficiency, as well as that of other features such as propensity for nucleosome positioning, will be important in future work.”

4) Towards excluding potential effect of mutations, the authors sequenced the strains around the relevant cis region. Could it be that the duplication detected at the URA3-mRuby cassette is a genetic change that confers advantage?

We agree with the reviewer that a duplication of the URA3-mRuby cassette would likely represent a genetic change conferring an advantage; however, as detailed in Supplementary file 1, such a duplication was only observed in one out of eight tuned lines that we sequenced from independent experiments. Our key finding from those experiments, though, was that in seven out of eight cases, tuning was observed without any observable beneficial genetic change. Thus, even if a duplication of URA3-mRuby could indeed eventually arise and be selected in the population once it began growing, such a duplication could not explain the initial emergence of growth. We have attempted to clarify this point in the revised manuscript by adding the following text:

“These data clearly indicate that the origin of growth-supporting URA3 expression levels in these cells cannot be reliant on a mutational mechanism, as only one of the eight cases – that with the URA3 duplication -- shows a mutation at high enough levels in the population to explain the onset of growth (mutations present in less than half of the population must have arisen after one or more cells in the population had already tuned and began growing, and thus by definition could not be responsible for the initial onset of the growing state)[…]

Our findings are consistent with a non-genetically heritable basis for the observed tuning in seven out of eight of the cases examined, as in all other growing lines, mutations near the URA3 gene were either non-existent or present only in a minority of the population.”